# Seasonal reconstructions coupling ice core data and an isotope enabled climate model – methodological implications of seasonality, climate modes and selection of proxy data

Jesper Sjolte[1], Florian Adolphi[1,3], Bo M. Vinther[4], Raimund Muscheler[1], Christophe Sturm[2], Martin Werner[5], and Gerrit Lohmann[5]

[1]Department of Geology – Quaternary Science, Lund University, Sölvegatan 12, 223 62, Lund, Sweden
[2]Swedish Meteorological and Hydrological Institute, 60176 Norrköping, Sweden
[3]Climate and Environmental Physics & Oeschger Centre for Climate Change Research, Physics Institute, University of Bern, Sidlerstrasse 5, CH-3012 Bern, Switzerland
[4]Physics of Ice, Climate and Earth, Niels Bohr Institute, University of Copenhagen, Denmark
[5]Alfred Wegener Institute, Helmholtz Centre for Polar and Marine Sciences, Bussestr. 24, 27515 Bremerhaven, Germany

**Correspondence:** Jesper Sjolte (jesper.sjolte@geol.lu.se)

**Abstract.** The research area of climate field reconstructions has developed strongly during the past 20 years, motivated by the need to understand the complex dynamics of the earth system in a changing climate. Climate field reconstructions aim to build a consistent gridded climate reconstruction of different variables, often from a range of climate proxies, using either statistical tools or a climate model to fill the gaps between the locations of the proxy data. Commonly, large scale climate field reconstructions covering more than 500 years are of annual resolution. In this method study we investigate the potential of seasonally resolved climate field reconstructions based on oxygen isotope records from Greenland ice cores and an isotope enabled climate model. Our analogue-type method matches modeled isotope patterns in Greenland precipitation to the patterns of ice core data from up to 14 ice core sites. In a second step the climate variables of the best matching model years are extracted, with the mean of the best matching years comprising the reconstruction. We test a range of climate reconstructions varying the definition of the seasons and the number of ice cores used. Our findings show that the optimal definition of the seasons depends on the variability of the target season. For winter, the vigorous variability is best captured when defining the season as December-February due to the dominance of large scale patterns. For summer, which has weaker variability, albeit more persistent in time, the variability is better captured using a longer season of May-Oct. Motivated by the scarcity of seasonal data we also test the use of annual data where the year is divided during summer, that is, not following the calendar year. This means that the winter variability is not split, and that the annual data then can be used to reconstruct the winter variability. In particularly when reconstructing the sea level pressure, and the corresponding main modes of variability, it is important to take seasonality into account, because of changes in the spatial patterns of the modes throughout the year. Targeting the annual mean sea level pressure for the reconstruction lowers the skill simply due to the seasonal geographical shift of the circulation modes. Our reconstructions based on ice core data also show skill for the North Atlantic sea surface temperatures, in particularly during winter for latitudes higher than $50^o$N. In addition, the main modes of the sea surface temperature variability are qualitatively captured by the reconstructions. When testing the skill of the reconstructions using 19 ice cores compared to the ones using 8

ice cores we do not find a clear advantage of using a larger data set. This could be due to a more even spatial distribution of the 8 ice cores. However, including European tree-ring data to further constrain the summer temperature reconstruction clearly improves the skill for this season, which otherwise is more difficult to capture than the winter season.

## 1  Introduction

Knowledge of past climate is essential to understand the range and processes of natural climate variability, impact of internal and external forcing, as well as serving as baseline to assess anthropogenic influences. The widespread implementation of weather observations dates back to about 1850, with sparse coverage in the early years. In order to investigate changes in weather and climate, as well as to evaluate climate models, so-called reanalysis data sets have been developed. Reanalysis data are gridded data products based on assimilation of weather observations using climate models. The use of reanalysis data sets has seen a wide range of applications due to the gridded data format and global coverage. However, due to being limited to the instrumental period, there is a strong incentive to develop similar products reaching further back in time.

In extratropical regions water stable isotopes from archives of paleo-precipitation are widely used as climate proxies for temperature, however the variability of water isotopes in precipitation is also related to atmospheric circulation. As vapor condensates from an air parcel during advection or ascent, the water molecules incorporating heavy isotopes ($^{18}$O, D) condensate more readily than lighter molecules. This means that the isotopic composition depends on the initial vapor content of the air parcel as well as its condensation history. The co-variability of vapor content and temperature results in the correlation between local temperature and the isotope composition of precipitation, while the dependency of the isotope content on the pathway results in the connection to atmospheric circulation. The $\delta$-notation is commonly used for the isotope ratio of a sample, giving the relative deviation from an isotopic standard (Craig, 1961).

Ice cores are some of the most important archives of the isotope composition of past precipitation. Some Greenland ice cores offer seasonal resolution, and in some cases even higher resolution (Furukawa et al., 2017), however the average annual ice accumulation must be larger than 0.2 m/year in order for the annual cycle of the isotope composition not to be completely smoothed out by diffusion in the firn (Johnsen, 1977). If the annual cycle is partly preserved it can be reconstructed by mathematical back-diffusion of the data. It has been shown that seasonal ice core data have high correlation to local temperature and circulation patterns, and that the summer and winter data reflect distinctly different spatial and temporal climate variability (Vinther et al., 2010). In particularly, Vinther et al. (2010) showed that the seasonal winter $\delta^{18}$O has better coherency with annual mean temperature than annual mean $\delta^{18}$O. This is due to weaker connection of the summer $\delta^{18}$O with summer temperature, and larger variability of both winter $\delta^{18}$O and temperature, which then dominates the annual signal Vinther et al. (2010). When studying climate further back than the earliest widespread weather observations, we rely on climate proxy data, such as ice cores. Inherent uncertainties in proxy data include age model uncertainties, seasonality, and if there is a stationary relationship between proxy and climate. This means that proxy data sets must be carefully chosen and evaluated, and the data must be well-studied to understand the relationship to climate before being incorporated in climate field reconstructions. Pioneering examples of climate fields reconstructions include Mann et al. (1998) who regressed climate patterns based on observations on

a collection of climate proxy data to obtain a global gridded data set of temperature, and Luterbacher et al. (2001, 2004) who reconstructed European sea level pressure and temperature with a similar regression technique, but also using early weather observations as well as historical documentation of weather variability.

Inspired by the techniques used for weather forecasts and reanalysis data, recent climate field reconstructions employ assimilation of climate proxy data using a climate model. The Last Millennium Reanalysis Project (LMR) (Hakim et al., 2016; Tardif et al., 2019) aims to make a global reanalysis using a wide range of proxy data. Their method includes proxy system modeling to link the proxies to the variables of the climate model. The regional studies of Sjolte et al. (2018) and Klein et al. (2019) are climate field reconstructions using Greenland and Antarctic ice core records, respectively. In the case of these two studies an isotope enabled climate model was used for the assimilation of isotope records from ice cores, which eliminates the step of calibrating the proxy records to a given environmental variable, such as temperature. These studies all use different statistical approaches when performing the assimilation procedure, where LMR employs a Kalman-filter, Klein et al. (2019) a particle trajectory approach and Sjolte et al. (2018) a variation of the analogue method, where the matching of model output to proxy data is done based on empirical orthogonal functions (EOFs). For a brief review of uses of the analogue method see Bothe and Zorita (2019). Common to the studies named above is the use of a static model ensemble. The latter means that there are no constraints on which model year can be chosen as analogy for a given year of the proxy data. This is mainly done for practical reasons since one avoids having to run ensemble simulations step-by-step as it is done for meteorological reanalysis data. One point that sets the study by Sjolte et al. (2018) apart from the other studies mentioned in this section is the use of seasonal proxy data in order to focus on reconstructing the winter season only, as opposed to targeting the variability of the annual mean. As mentioned above in connection with the study by Vinther et al. (2010), the Greenland ice core data show distinctly different variability between summer and winter. Such differences in variability may originate in the relation between climate proxy records and climate variability, for example due to different climate sensitivity through the seasons, or due to climate variability itself, for example the change of circulation regimes during the year (Hurrell et al., 2003). Due to these questions of seasonality climate field reconstructions targeting the annual mean could therefore, by the nature of both the climate proxies and climate variability, have limited skill. This could to a large extent depend on the definition of the year, and may bias reconstructions towards specific seasons despite the use of annual data. The issue with seasonality could in particularly play a role when it comes to atmospheric circulation regimes, which shall return to later in Section 4.1.

In this study we will investigate the methodological implication of extracting seasonal and annual climate information from Greenland ice cores using a coupled model-data approach. We will use the method by Sjolte et al. (2018) with an extended data set including summer and annual isotope data from ice cores, as well as tree ring chronologies from Europe. In combining model output with these data sets, we reconstruct sea level pressure (SLP), surface air temperature (T2m) and sea surface temperature (SST). We will test

- The influence the number of ice cores assimilated for the reconstruction

- If the definition of the seasons impact the skill and recorded climate variability in the reconstructions

- If annual data can be used to reconstruct winter variability

– To which extent the governing atmospheric circulation modes can be reconstructed using summer, winter and annual
data

  – If including tree ring data can improve the reconstruction for the summer season

  – If the reconstructions capture variations in the North Atlantic SSTs, hereunder the main modes of the SST variability

## 2   Data

In this study we use the seasonal $\delta^{18}O$ ice core data of Vinther et al. (2010). We use the data for summer (May-Oct), winter
(Nov-Apr) and winter centered annual mean (Aug-Jul) as defined by Vinther et al. (2010). To achieve the longest possible data
set with the best regional coverage we chose 8 cores covering 1241-1970, and for the largest data set possible we chose all 19
cores covering 1777-1970 (Supplementary Figure S1).

In addition to using the ice core data, we produce reconstructions for summer where tree-ring data is used to further constrain
temperature. Tree-ring chronologies using primarily maximum late wood density as climate proxy can have a strong sensitivity
to summer temperature. Such records are compiled in Wilson et al. (2016). From this compilation we select tree-ring records
that cover the entire study period (1241-1970), and correlate well with local temperature. This leaves us with 8 tree-ring records
from Europe (Table 1).

We use the isotope enabled version of ECHAM5/MPI-OM (Werner et al., 2016) in T31L19 configuration, which corresponds
to $3.75^{o}$ x $3.75^{o}$ horizontal resolution using 19 vertical hybrid levels. The model includes isotope tracers in a fully coupled
hydrological cycle, with fractionation taken into account for all phase transitions. The simulation covers year 800-2005 with
natural and anthropogenic forcings, including greenhouse gases, volcanic aerosols, total solar irradiance, land use and orbital
forcing. See Sjolte et al. (2018) for full details on the model run.

To evaluate the skill of the reconstructions we use the 20th Century Reanalysis Version 2c (20CR) (Compo et al., 2011) for
the period 1851-1970, as well as the accompanying COBE SST data (Ishii et al., 2005). We mainly use 20CR to assess the
110 skill in for spatial correlation patterns and assessing modes of variability. 20CR has well known biases (Reeves Eyre and Zeng,
2017) and care should be taken when performing detailed analysis using this data set. In addition to the evaluation using 20CR
we compare the reconstructions to the south west Greenland temperature data compiled by Vinther et al. (2006), which is
continuous 1874-1970, as well as data from Stykkisholmur, Iceland, which covers 1830-1970 (Jónsson, 1989). These data are
the longest running instrumental temperature data available relatively close to the ice core sites used here. Finally, the station-
115 based record of the North Atlantic Oscillation (NAO) by Jones et al. (1997) is used for evaluating the reconstructed NAO for
the period 1824-1970.

We follow the convention of using the term principal components (PCs) for the time series of the main modes of variability,
while using the term EOFs for the spatial patterns of the modes. The method of Ebisuzaki (1997) is used to calculate the
significance when correlating filtered time series in order to take auto-correlation into account.

 **3  Methods**

### 3.1  Selection of model analogues based on ice core data

We use the reconstruction method of Sjolte et al. (2018) to produce a number of reconstructions of different length, different definitions of the seasons as well as varying the number of proxy records in the data set. The reconstruction method can be classified as assimilation of proxy data using the analogue method with a fixed model ensemble. This method identifies analogues, i.e. years, in a climate model simulation most closely matching the annual or season spatial pattern in a set of proxy data. In order to capture the characteristic regional variability of Greenland $\delta^{18}O$, and to smooth out the noisy signal of individual ice cores, the matching of the model output is done using EOFs. Conventionally, proxy data needs to be calibrated to a given climate variable, e.g. temperature, in order to be compared to a climate model. The use of an isotope enabled climate model makes it possible match the proxy data with modeled patterns without calibration, since the proxy itself is included in the model output. This important feature of the method means that we include the governing processes of the variability in the proxy data, capturing the integrative nature of isotope proxies and the information that lies therein (see introduction). The work flow of the reconstruction is to i) calculate the PCs from the respective covariance matrix of the ice core $\delta^{18}O$ ($PC_{icecore}$) and modeled $\delta^{18}O$ ($PC_{model}$) retaining the first three PCs, and evaluate the modeled patterns for a given model year ($t'$) against the ice core patterns (Figure 1) for each proxy year ($t$) using Eq. 1 ii) sort the model simulation by comparing the isotope patterns each year of the model simulation to the isotope patterns each year of the ice core data, using the normalized PCs to achieve equal weighting for the regional variability iii) define the best matching model years as ensemble member one, the second best matching years as ensemble member two, and so on, and test how many ensemble members to retain (p<0.01) by calculating the Chi-square statistic between the modeled and the ice core PCs iv) extract the climate field variables from the selected model ensemble and calculate the ensemble mean, which comprises the climate reconstruction.

$$\chi^2_{Match-IC}(t) = \frac{1}{3}\sum_{k=1}^{3}(PC(k,t')_{model} - PC(k,t)_{icecore})^2 \tag{1}$$

The number of ensemble members (see Table 2) depends on the degrees of freedom, i.e., the length of the reconstruction, and how many closely matched model analogues that are found. In order to assess the quality of the matching exercise we extract the ensemble mean reconstructed $\delta^{18}O$ at the ice core sites and correlate it against the ice core $\delta^{18}O$. This tests if matching the modeled PCs to the ice core PCs captures the variability of the original ice core data. The performance is similar for summer, winter and annual data, and the signal of the ice core data is well captured, with correlations ranging from 0.4 to 0.8 (Supplementary Figure 2). The highest correlations are seen for sites with multiple ice cores and high accumulation rate, both of which reduces noise. In summary, the reconstructed $\delta^{18}O$ captures the regional variability of of the ice core data well based on matching the normalized $PC_{model}$ and $PC_{icecore}$.

As outlined in the introduction the definition of the seasons or year is an important parameter for the reconstruction. This applies both in terms of the seasonality of the proxy data and the target season of the reconstruction. Following the study of Vinther et al. (2010) we will use the definitions of summer as May-Oct (sum50), winter as Nov-Apr (win50), and winter

centered annual mean Aug-Jul (win100) for the ice core data. These definitions will also be applied to the target seasons of the reconstructions, as well as the widely used definitions of summer (JJA) and winter (DJF). We investigate the seasonal and annual variability using these different definitions with two data sets for *short* (1777-1970, 19 ice cores) and *long* (1241-1970, 8 ice cores) reconstructions, resulting in a total of 12 reconstructions, where one for DJF covering 1241-1970 was published by Sjolte et al. (2018) (see Table 2).

## 3.2  Constraining summer reconstructions using tree-ring data

For the summer season we test incorporating tree-ring data to further constrain the reconstruction. We choose a simple approach of incorporating the data, which can serve as a pilot study for further tests of adding more data to the reconstruction. For the test we sort the pre-selected 39 ensemble members ($t'_{IC-ENS}$) based on the ice core data (Table 2) using a Chi-square fit of normalized modeled temperature at the 8 tree ring sites ($T_{model}$) against the normalized tree ring data ($T_{trees}$) (see Eq. 2).

$$\chi^2_{Match-TR}(t) = \frac{1}{8} \sum_{k=1}^{8} (T(k, t'_{IC-ENS})_{model} - T(k, t)_{trees})^2 \tag{2}$$

The fit is done using the JJA temperature from the model, which are the best months to use with respect to seasonal sensitivity for these 8 tree-ring records (Wilson et al., 2016). In a next step we test the ensemble mean temperature reconstruction against the time series of the tree-ring data at each site, by calculating the correlation to the tree-ring data while increasing the number of ensemble members from 1 to 39 (Supplementary Figure S3). Although a Chi-square test of the fit of the reconstructed temperature shows that including 24 ensemble members provides a good fit (p < 0.01), the correlation decreases quite rapidly when including more ensemble members and we choose to include only 20. With this ensemble we capture the variability of the tree-ring data relatively well for the whole period of the reconstruction (Supplementary Figure S4). The correlation goes to zero when including all of the 39 ensemble members, indicating that without the tree-ring data the reconstruction using only the 8 ice cores and the model has no predictive skill of the summer temperature in Europe.

## 4  Results

### 4.1  The seasonal variability in observations and when combining proxy data and model output

In the introduction we mentioned seasonality, definition of seasons and shifts in circulation patterns as potential limiting factors for the skill of climate field reconstructions. In general, seasonal dependency on climate variables, temporal resolution as well as the precision of the chronology of proxy records sets a limit on the temporal resolution of climate field reconstructions. Seasonal resolution is likely the highest possible resolution which can be attained due to these different factors. The sub-seasonal auto-correlation structure of atmospheric variability is a key factor in how well seasonal proxy data can represent climate variability. This can be illustrated by investigating the monthly auto-correlation during the year of the 1st leading mode

of sea level pressure in the North Atlantic region, the NAO. We found that, for example, the 2nd and 3rd leading modes are too dissimilar between summer, autumn, winter and spring to allow a meaningful study of the monthly auto-correlation of these modes, as they simply represent different teleconnection patterns during each season. Figure 2 shows the monthly auto-correlation of each month of the PC-based NAO calculated from the 20CR. These figures show that during the cold season the NAO has the weakest auto-correlation with other months, as well as weaker year-to-year auto-correlation compared to summer. While the lower auto-correlation during winter shows stochastic nature of the variability, it is also during winter that the NAO variability is the most vigorous. Thus, the portion of a given climate signal that can be reconstructed is a balance of what is recorded in the proxy at a certain resolution, as well as the strength and auto-correlation of the signal sampled at this resolution. It is noteworthy that Figure 2 also illustrates that targeting the calendar year in a reconstruction (or any sort of analysis) splits up the variability mid winter and mixes the variability of two consecutive winters that have little variability in common. This is the motivation for using the definition of winter centered annual mean for the annual data in this study.

Vinther et al. (2010) tested the ice core data used in this study using correlation with observed temperature, leading to the division of the in seasons using the definition of sum50, win50 and win100 as outlined in Section 3. Due to the changes in the patterns and variability of the circulation modes from summer to winter we furthermore test the seasonality in terms of circulation modes. We do this by performing monthly reconstructions for pressure and correlating the time series of the corresponding main modes of circulation against that of the modes of the 20CR. This is done using the same method as for the seasonal reconstructions, but only picking individual months from the matching year of the model simulation. We do not suggest that it is feasible to reconstruct climate on monthly timescales using seasonal ice core data. This exercise is purely for testing purposes. The monthly reconstructions are done for each data set (sum50, win50, win100, for 8 ice cores and 19 ice cores) for the months that each data set is assumed to cover, e.g. May-Oct for sum50. The overall results show that the different reconstructed surface pressure modes, as represented by the first three PCs, do not peak in skill during the same months (Supplementary Figures S5 and S6). For example, for win50 PC1 has highest skill for Feb-Apr, while the skill for PC2 peaks Jan-Feb. This type of behavior is repeated for the sum50 and win100 data sets. The differentiated seasonality in the skill of the reconstructed modes can originate from i) the sensitivity of the Greenland $\delta^{18}O$ to different modes ii) the changes in circulation modes during the season iii) the auto-correlation structure of circulation, as discussed above iv) model biases in circulation modes, and combinations of these influences. The difference in the reconstructions using 8 ice cores and 19 ice cores, respectively, is mainly seen for win100, where more monthly reconstructions show significant skill across the year when using more ice cores in the reconstruction. Furthermore, the monthly skill for the win100 data set indicate that it is feasible to reconstruct the winter circulation (e.g. DJF). This test suggests that in order to get the highest average skill possible for all modes during winter the reconstruction should target DJF, while for summer the full span of the season (May-Oct) is likely better, also taking into account the higher monthly auto-correlation during the warm season. The EOF patterns of surface pressure will be discussed further in Section 4.2.2.

## 4.2 Evaluation of reconstructions

In the following sections we evaluate and compare the reconstructions using different methods. We start with point-by-point correlation maps for the North Atlantic sector of the reconstructions to 20CR SLP and T2m as well as the COBE SSTs. This is a general evaluation in terms of spatial coverage and skill of the reconstructions. We also include a comparison to the longest instrumental records of temperature from Greenland and Iceland. Next we evaluate the skill of the reconstructions in terms of atmospheric circulation modes. In the final part of the evaluation we investigate to which extent the main patterns of North Atlantic SSTs and their variability can be reconstructed using the method of this study. We would like to emphasize that none of these reconstructions have been calibrated to observations. Instead, the model provides us directly with the physical variables of SLP, T2m and SST from the model years where modeled and measured $\delta^{18}$O patterns match. The evaluation of these reconstructions are thus done using completely independent data sets.

### 4.2.1 Reconstructed temperature and sea level pressure

Investigating the results for correlations and the spatial patterns of skill for SLP, T2m and SSTs reveals a complex interplay of factors influencing the reconstructions for different seasons, as well as how the different definition of seasons influence the skill. Reconstructions for the summer season show the least skill, but perform better using the extended definition of the target season (May-Oct) (Figure 4) rather than JJA (Figure 3). The summer reconstruction also appears to benefit the most from including 19 ice cores rather than 8. This can be seen also be seen from Table 3, which summarizes the maximum correlation and number of significantly correlated grid points compared to 20CR for all reconstructions in this study. Including more cores and using the extended season likely reduces noise in the reconstruction. Using the extended season also smooths out the variability of the 20CR data, which can partly account for the higher skill of the short sum50 reconstruction for summer. The summer reconstructions using 8 ice cores show no significant skill for Europe, which is in line with the correlation analysis with European tree-ring data (see Section 3.2). However, the evaluation of the summer reconstructions using 19 ice cores shows patches of significant correlation in Europe. The reconstructions for winter shows the highest skill of the reconstructions, in-line with the findings of Vinther et al. (2010), that $\delta^{18}$O is found to be a more efficient climate proxy during winter (Sjolte et al., 2011, 2014). This can partly be due to the climate variability in extra-tropical North Atlantic region being most vigorous during winter causing a large signal-to-noise ratio in $\delta^{18}$O records with respect to their ability to record circulation changes. All of the these factors contribute to better reconstructions for winter compared to summer, both in terms of spatial skill and strength of correlation with 20CR. This includes significant temperature skill in Northern Europe, which is probably due to the reconstruction capturing the main modes of SLP. We will return to this topic in Section 4.2.2. As opposed to summer, the winter reconstructions for DJF performs better, rather than the extended season Nov-Apr. This is probably due to the migration of circulation patters and low auto-correlation of atmospheric circulation during winter as discussed in Section 4.1.

One of the questions of this study is about the use of annual data for reconstructions of climate and atmospheric circulation. For the reconstructions targeting the winter centered annual mean (win100) the skill and patterns of correlation are reminiscent to that of the winter reconstructions, although clearly with less areal coverage of significant correlation for SLP. We interpret

this as being due the migration of the circulation patterns with the seasons, as discussed above. However, for SSTs the win100 reconstruction shows the highest spatial skill of all the reconstructions, including better capturing low latitude variability, with
250 the correlation pattern being reminiscent of the spatial pattern of Atlantic Multi-decadal Oscillation (AMO) -type variability. As with the extended summer season, part of the increase in skill for the win100 SST reconstruction could also originate from a smoother signal for annual data – in both observations and reconstruction, where some of the noise is reduced compared to seasonal data, but some of the signal is also lost. Targeting the winter season (DJF) using the winter centered annual data results in a clear gain in skill for SLP, while the skill for SST is somewhat reduced, although retaining the overall correlation pattern of
255 the winter centered annual mean reconstruction. This indicates that it is feasible to reconstruct winter variability from annual data, if the definition of the winter centered annual mean is used for the proxy data. Seasonal $\delta^{18}O$ data are increasingly sparse going back in time, and using winter centered annual mean data could be an alternative for reconstructing winter variability beyond the reach of seasonal $\delta^{18}O$ data when seasonality in the ice can still be defined from e.g., aerosol records.

To further assess the skill of the reconstructed temperature we compare to data from three stations on the Greenland coast
and one Icelandic station. Vinther et al. (2010) showed that the first Principal Component (PC1) of Greenland isotope data (20 cores) has strong correlation (r = 0.71) to the stacked Greenland coastal data (South West Greenland temperature, SWG index) during winter (Nov-Apr), while PC1 of the isotope data for summer is most strongly correlated to data from Iceland (r = 0.55) (May-Oct). Here we compare the reconstructed site temperature both to data from each of the stations and to the SWG index. The highest correlations are found for the 8 core Win50 reconstruction at Nuuk and Qaqortoq with a correlation
of 0.6 at both sites (Figure 5 and Table 4). It is also for this reconstruction we find the highest correlation of 0.63 with the SWG index. While the correlation for Ilulissat is similar to the correlation for Nuuk and Qaqortoq, the observed higher amplitude is not captured by the reconstruction, which is probably due to subgrid variability neither resolved by reconstruction nor the model. The 19 core reconstructions have slightly lower correlations to the Greenland temperature data. This could be due to a weighting of the variability more to the east, as most of the additional cores in the shorter reconstructions are to the east of
the ice divide. For the summer reconstructions the correlations to the Greenland station data are below 0.3. However, the 8 core Sum50 reconstruction captures a substantial part of the longer term variability with a correlation of 0.44 to the decadally filtered SWG index. With respect to the definition of the winter season, the DJF reconstructions appear to better capture the long term variability with slightly higher correlation for the filtered data compared to the Win50 reconstructions. The Win100 and the Win100 DJF reconstructions both show only slightly lower correlations than the Win50 and DJF reconstructions, indi-
cating that for temperature alone the seasonal data is less crucial than for reconstruction SLP, at least when comparing locally to the Greenland coastal data.

The correlations to the Icelandic temperature data shows correlations around 0.3 for all reconstructions, with most of the summer reconstructions showing higher correlations for long term variability compared to the winter reconstructions. This indicates a similar behavior as for the ice core PC1 correlation with respect to the winter data responding more to the Western
Greenland temperature and the summer data having better coherency with Icelandic data. The predominance of the summer signal east of Greenland also results in the reconstructions based on the winter centered annual mean not having very high skill for Icelandic temperatures, at least for the long term variability.

Comparing the summer reconstructions including tree-ring data with the 20CR we find that the skill for SLP, T2m and SST has increased considerably compared to the summer reconstructions only using 8 ice cores (Table 3 and Supplementary Figure S9). The skill is improved in particularly for temperature in the eastern sector of the domain, while the skill for SLP is still low near Greenland, although the skill has clearly increased over Northern Europe for JJA.

### 4.2.2  Main modes of atmospheric variability

Sjolte et al. (2018) showed that the winter variability of the first two PCs of the SLP in the North Atlantic region could be reconstructed with good skill using the analogue method based on 8 ice cores. Here we evaluate all the different reconstructions of this study for the first three PCs, including the spatial patterns of the loading of the PCs (EOFs). For the DJF and Win50 reconstructions EOF1, 2 and 3 all qualitative match that of the 20CR (Figure 6). The reconstructed EOF patterns for SLP are very similar for the reconstructions using 8 and 19 ice cores, respectively, and we only show the patterns for the reconstructions using 8 ice cores. There are some indications that EOF2 of the reconstructions summarizes some of the variability assigned to EOF3 of the 20CR as also discussed by Sjolte et al. (2018). For summer the reconstructed EOFs capture many of the same features of the 20CR, but less clear than for the winter reconstructions. For example, the reconstructed JJA pattern for EOF1 shows differences to 20CR south of Greenland (Figure 6), which probably partly explains the low skill for summer SLP in this region shown in Section 4.2.1. The origin of this problem is probably a bias for large scale summer variability of the ECHAM5/MPIOM model (Jungclaus et al., 2006). This means that the main modes of the original model simulation (not shown) do not correspond to the main observed modes, except for Winter NAO, which the model captures. It is only after matching the model output to the proxy data that the main modes align with the observed patterns.

The maps of the EOF patterns illustrate the point made earlier about the differences in the modes of SLP variability from season to season. Not only do the patterns change from summer to winter but also depending of the definition of the season, e.g. JJA versus May-Oct (Supplementary Figure S11). Furthermore, the EOFs of the winter centered annual mean appear as mixtures of summer and winter variability, carrying most likeness to the winter patterns, again showing the problem of using the annual mean SLP as target for reconstructions.

Common for all the different reconstructions is that they all assign more variability to EOF1 and less to EOF3 compared to 20CR, while EOF2 is fairly similar to 20CR in terms of the explained variance. This could be due to sole use of Greenland ice core data, which could skew variability to be dominated more by NAO-type variability. For DJF the model simulation itself does not have a high bias in the explained variance of NAO-type variability.

From the time series of EOFs (PCs) it is evident that the reconstructions have realistic amplitudes of the year-to-year variability (Figure 7). In other words, the spectrum of the reconstructions are similar to actual weather variability as also found for the DJF reconstruction by Sjolte et al. (2018). Correlating the reconstructed PCs to that of the 20CR (see Figure 8) shows that i) the variability of PC1 is well captured by the winter and annual data ii) only the Win50 DJF reconstruction has skill for PC2 iii) the summer reconstructions have some skill for PC3 iv) in some instances the decadally filtered data capture a significant part of the 20CR variability, even with no correlation for annual data (e.g. PC2 and PC3 of DJF Win100 (8 cores)). The very

low values 1851-1860 in the 20CR PC1 is possibly a bias in the reanalysis and is not seen in the HadCRU NAO time series (not shown). Comparing the reconstructions for winter and annual data to the HadCRU NAO results in higher correlations than for 20CR, also for the filtered data. For summer it is not meaningful to use the station-based NAO due to the shifted centers of action during summer compared to winter. As discussed in Section 4.2.1 the skill for SLP improves locally when including tree-ring data to constrain the summer reconstructions. However, the skill for the circulation patterns is not improved by including the tree-ring data.

### 4.2.3   North Atlantic sea surface temperature

The correlation maps with the COBE SSTs (Figures 3 and 4) indicate that the reconstructions are particularly well suited to investigate the SST variations in the region $50^oN$-$70^oN$, $70^oW$-$0^oW$. For this purpose we define a North Atlantic SST index as the mean SST for the aforementioned area. Although the year-to-year variations of the reconstructions are somewhat noisy compared to the variations of the COBE SSTs, the reconstructions have significant skill for all investigated seasons, most notably for winter and annual data (Figure 9). For decadally filtered data the Win50 DJF and Win50 reconstructions (8 cores) explain more than 50% of the COBE North Atlantic SST variability (r = 0.72 and r = 0.74, respectively) (Figure 10). While the long term SST changes for summer are underestimated, the reconstructions of winter SST match the COBE amplitudes of the decadal-multidecadal SST variability very well. As mentioned in Section 4.2.1 the skill for temperature and SST is markedly improved when including the tree-ring data in the summer reconstructions. This is also see in the higher correlations and stronger significance for the North Atlantic SST index for these reconstructions (Figure 10).

To further investigate how much information of the North Atlantic SST variability is obtainable using this type of reconstruction, we also compared the patterns and variability of the main modes of reconstructed SSTs to that of the COBE SSTs (Figures 11). As the skill of the reconstructions decreases with the distance from the proxy sites we calculated the modes using data from $30^oN$-$70^oN$ for the reconstructions, while we used $0^oN$-$70^oN$ for the COBE data. Generally the reconstructions qualitatively capture the spatial characteristics of the EOF1, 2 and 3 patterns of the COBE data, as well as the variability of the PCs (Figure 12). Again, the match appears to be better for the winter season. The PCs of the reconstructed SSTs are correlated to the reconstructed PCs of SLP, indicating that the SST variability captured by the reconstruction is related to atmosphere-ocean interaction of the main circulation modes (not shown). EOF1 of the SSTs is also correlated to the North Atlantic SST index discussed above, and the pattern is akin to AMO-type variability associated with long term variation of the NAO (McCarthy et al., 2015). EOF2 of the SSTs can be related to subpolar gyre-type variability connected with the frequency of the weather patterns Atlantic Ridge/Blocking (Moffa-Sanchez et al., 2014; Moreno-Chamarro et al., 2017). Only the reconstructed PC1 for winter and annual SSTs shows consistent skill compared to the COBE SSTs, although the Win50 PC3 (19 cores) also has significant correlation for both annual and decadally filtered data (Figure 13).

### 4.3   Comparison to other millennium-length reconstructions

While an exhaustive comparison to other reconstructions is beyond the scope of the this study, we briefly compare our reconstructions to two other data sets. We limit ourselves to reconstructions that are based on data entirely independent from this

study and also cover the span of our longest reconstructions (1241-1970). We first compare to the temperature index for Central Europe by Glaser and Riemann (2009), which is based entirely on historical documentation and early instrumental data. Due to less available data in the early part of the millennium, the reconstruction by Glaser and Riemann (2009) is only in seasonal resolution prior to 1500 CE, while monthly data is available after this. Due to this change in resolution and variability we only show the comparison to Glaser and Riemann (2009) for the period after 1500 CE. In the comparison we use our reconstruc-

tions including tree-ring data for summer (JJA, sum50), as the reconstructions relying solely on ice core data (8 ice cores) do not have skill in Europe for summer. Judging from the moving correlation there is fairly good correspondence between our reconstructions and the temperature index of Glaser and Riemann (2009) for the period after 1600 CE, apart from a distinct spell of out-of-phase variability around 1650 CE for the summer season (Figure 14). The correlation is most consistent for DJF, although the decadal to multidecadal variability also appears coherent for the summer season. For the period prior to 1500 CE

(no shown) little coherency is seen between our reconstructions and the temperature index of Glaser and Riemann (2009). As the temperature index of Glaser and Riemann (2009) relies on a relatively few data for the early part of their reconstruction it is tempting to conclude that the loss of correlation is due to this, as our reconstructions is produced with the same number of records and same method throughout the reconstructions. Despite this, the comparison provides support of the validity of our seasonal temperature reconstructions extending further back than the comparison to reanalysis data.

In a second comparison we include the recent DJF NAO reconstruction by Cook et al. (2019) which is based on drought data from tree rings. For reference we also include the comparison to the model constrained NAO reconstruction by Ortega et al. (2015) also shown in Sjolte et al. (2018), although this reconstruction is also partly based on Greenland ice core data. From the moving correlation there is little correspondence between our NAO reconstruction and that of Cook et al. (2019) prior to the instrumental record (Figure 15). Unlike our method, the method of Cook et al. (2019) involves calibration to observed the

NAO. Also for the decadal to multidecadal time scales the variability of the reconstructions diverge prior to the instrumental record, including the reconstruction by Ortega et al. (2015). This indicates that the long standing problem of incoherence between different NAO reconstructions prior to the instrumental record is still valid (Pinto and Raible, 2012). The reconstructions shown in Figure 15 b)-c) are scaled to the decadal variability of the observed NAO to facilitate comparing the interannual variability. It is clear that the interannual amplitude of the reconstruction by Ortega et al. (2015) is underestimated, while our

reconstruction appears to be only slightly underestimated in amplitude, and the reconstruction by Cook et al. (2019) could have a somewhat overestimated interannual variability. Factors which could contribute to the lack of correlation between our and the reconstruction by Cook et al. (2019), is that the relationship between drought and winter NAO is not stationary in time (López-Moreno and Vicente-Serrano, 2008), and that the number of records in the reconstruction by Cook et al. (2019) decrease strongly back in time prior to 1700 CE.

**5  Discussion and conclusions**

In this study we tested climate reconstructions of summer, winter and annual climate variability, based on a data set of 8 ice cores covering 1241-1970 and an extended data set of 19 cores covering 1777-1970. While the increased number of ice cores

can reduce noise in the reconstructions, the more geographically uneven distribution of the additional cores appears to have some negative effects for the skill of the reconstructions. This means that the over all added value of more ice core data seems

less than the drawbacks of the much shorter time span being covered. Unfortunately it is not possible to test the reconstructions of 8 versus 19 cores on truly equal terms, as the EOFs of the 8 ice cores for shorter time periods are dependent on the exact choice of the investigated time period. This is due to poor statistics in determining the EOFs when the number of ice cores is low and the data sample is short.

The inherent properties of climate variability with respect to auto-correlation and changes in governing weather patterns as

illustrated in Section 4.1 are probably the reasons for the differences in skill seen for the reconstructions using different definitions of the target season. One consequence is that the skill for secondary circulation modes is better for the reconstructions targeting DJF rather than Nov-Apr, and a secondly that using the wider definition of summer (May-Oct) may reduce some noise in the temperature reconstruction, an effect which likely also can be seen for the temperature reconstructions of the winter centered annual mean. Additionally, reconstruction of the DJF atmospheric circulation using winter centered annual

mean ice core data is attainable, which opens up the possibility of extending the winter reconstructions further back than with seasonal data. This could be done by using high resolution chemistry data (e.g., Rasmussen et al., 2006) to define the seasons in the ice core data, even though the annual cycle in the ice core isotope data cannot be recovered.

The evaluation of correlation to the North Atlantic SSTs shows a particular strong sensitivity to SSTs variability north of $50^o$N. This is in principle true for all seasons, but in particular in winter, where the amplitude of the decadal changes in SSTs

are captured by the reconstruction. This is achieved without tuning the reconstruction to observations. This indicates a clear potential for reconstructing AMO-like variability. Furthermore, the reconstructions yield qualitatively similar main patterns of variability as those based on observations (EOF1, 2 and 3). These SST patterns are connected to the main atmospheric modes of variability.

The reconstructions in this study only based on ice core data are using what one might call a minimal proxy data set. The

405 thought behind is to select few – but high quality, well dated, and well studied proxy data, rather than a large collection of data where the link between climate parameters and all proxy data has not been tested in details. Furthermore, the use of isotope records have the property discussed in the introduction of not only recording local information, while the assimilation using an isotope enabled climate model allows coupling the model and proxy data without calibration. However, it is clear that the skill of the summer reconstructions is generally lower than the the winter reconstructions. For this reason we also perform a

410 test including European tree-ring data for two additional reconstructions for summer (JJA and May-Oct) covering 1241-1970. For these reconstructions the skill for temperature is clearly improved, although for SLP the skill only improves locally with no improvement of the skill for the main modes of circulation.

For model assimilation-type climate reconstructions the performance of the climate model is an important parameter. All climate models have biases that can influence the patterns of the reconstructed climate variability. Here we have mainly discussed

the model bias in SLP during summer as this is the most prominent model related problem found for our reconstructions. Given the relatively coarse model resolution ($3.75^o$ x $3.75^o$) using a model with finer resolution, better representation of orography, atmospheric circulation and physics would probably yield a better climate reconstruction. However, the model used in this

study fundamentally performs well when it comes to mimicking the variability of the isotopic composition of Greenland precipitation, which is what allows us to use the method of matching the ice core EOF patterns.

Different strategies can be chosen for attaining an uncertainty estimate of the reconstructions based on the analogue method. Bothe and Zorita (2019) presents different options i) uncertainty based on the fit of the analogues to the proxy data ii) a fixed distance allowed for the fit of the analogues, but variable number of analogues, and iii) uncertainty estimated from the ensemble spread of model analogues. Our method employs a fixed number of model analogues (e.g. 39 for DJF 1241-1970) and the ensemble spread is therefore the most natural choice of uncertainty estimate. When comparing to other data sets the RMSE can

also be used along with the correlation coefficient as a measure of how well reconstruction matches the variability. This can for example reveal cases where the correlation is good, but the amplitude of the variability does not match (see Supplementary Figure S16). In Figure 5, where we plot time series of Greenland coastal temperature, we both show the ensemble spread and the RMSE with respect to the observations. Except for Illulissat, which has very high observed variability, the ensemble spread and RMSE is very similar. This indicates that the ensemble spread is a good measure of the uncertainty at a grid point

scale. In the comparison to other NAO reconstructions we also show the ensemble spread and the RMSE with respect to the observations (Figure 15). In this case the RMSE is well within the envelope of the ensemble spread of our reconstructed NAO, indicating that the spread is a relatively conservative measure of uncertainty. In addition we have investigated the quality of the fit over time (Chi-square distance for each time step) to see if there are trends or periods of very poorly fitting model analogues. Although there are years where we have trouble finding a good model analogue, the fit is on average throughout the records

as good as for 1851-1970 where the reconstructions are evaluated. For example, there are no large decadal trends in the fit. From a statistical point of view, the reconstructions are therefore equally valid any time during the reconstruction as there is no calibration involved in the method.

The approach of using an ensemble of analogues improves the reconstruction in terms of correlation to observations, but also reduces the variability when producing the ensemble mean due to averaging out some of the variability (Gómez-Navarro et al.,

2017). Using the example of the Greenland coastal temperature again (Figure 5), the amplitude of the year-to-year variability is somewhat underestimated in the reconstruction, while the decadal-scale variability is well captured. This smoothing of the high frequency variability in the reconstruction can to a certain extent be attributed to the ensemble approach, but also to the relatively course resolution of the model, which also smooths out variability. On the other hand the SST reconstruction (Figure 9) shows an overestimated variability for winter, which could be due to using an atmospheric signal to reconstruct ocean

variability, while the amplitude is underestimated for summer. This contrast can probably be explained by the lower skill for summer, which causes loss of variability due to lack of coherency in the ensemble. For the reconstruction of atmospheric circulation (SLP), the amplitude year-to-year variability is well preserved and the ensemble averaging appears to have a minor effect on the high frequency variability (Figure 8 and Figure 15). One factor in preserving the year-to-year atmospheric variability, is that we are sampling from a model simulation where, for example, the NAO has a nearly white power spectrum (not shown)

and given that the ensemble spread is relatively large (Figure 15), this spectrum will be preserved in the reconstruction.

To attain the best possible reconstruction of climate variability, taking into account the the nature of the target for the reconstruction is important. This is illustrated by the dependency of the skill of the climate reconstructions on the definition

of seasonality, due to the seasonal changes of the patterns or variability. For winter a narrow definition of the season (DJF) yields better performance for circulation patterns. Furthermore, in some cases a wider definition of the season, e.g. for summer and annual data, can provide better performance for temperature due to better capturing the signal during months of higher auto-correlation and less variability.

Further development of seasonal climate field reconstructions requires a larger data set of well studies proxy records. Isotope records of tree-ring cellulose from regions with sustained winter snow are potential sources for expanding the spatial coverage for winter (Seftigen et al., 2011; Edwards et al., 2017). In more temperate climates such records could be used for reconstructing summer variability (Labuhn et al., 2016). Speleothem data could potentially also be used, however is a challenge to find high resolution continuous data sets due to growth hiatuses (e.g., de Jong et al., 2013). Newly updated isotope enabled climate models (e.g., Cauquoin et al., 2019) shows the continual development of this field. This makes running new millennium length model simulations attractive for the purpose of providing better sampling pools for finding model analogues to match the proxy data. Although not shown in this study, reconstruction of precipitation is also possible using the analogue method. However, in particular for precipitation better model resolution is important to capture storm tracks and orographic effects. Finally, the indication found in this study of that is possible to capture the main SST patterns of the North Atlantic, makes this approach a good supplement to marine records due to better precision of the dating of terrestrial records.

*Code and data availability.*  The code for the reconstruction method as well as the reconstructions shown in this paper are available upon request to the corresponding author. The time series for PC1 and PC2 of reconstructed DJF SLP in the North Atlantic Region previously published by Sjolte et al. (2018) are available from the PANGAEA open access data library (https://doi.pangaea.de/10.1594/PANGAEA.892841).

*Author contributions.*  J.S. developed the method, did the reconstructions, performed the analysis, and wrote the first version of the manuscript. F.A. contributed to the writing, method development and analysis. B.V. provided seasonal and annual ice core data. R.M. contributed to method development. C.S. contributed to the initial study and method development. M.V. and G.L. provided technical support and insights on climate modeling. All authors discussed and edited the manuscript.

*Competing interests.*  The authors declare no competing interests.

*Acknowledgements.*  This work was supported by the Swedish Research Council (grant DNR2011-5418 & DNR2013-8421 to R.M.), the Crafoord foundation and the strategic research program of ModEling the Regional and Global Earth system (MERGE) hosted by the Faculty of Science at Lund University. F.A. was supported by the Swedish Research Council (grant DNR 2016-00218 to F.A.). Support for the Twentieth Century Reanalysis Project version 2c dataset is provided by the U.S. Department of Energy, Office of Science Biological and

Environmental Research (BER), and by the National Oceanic and Atmospheric Administration Climate Program Office. COBE SST data provided by the NOAA/OAR/ESRL PSD, Boulder, Colorado, USA, from their Web site at https://www.esrl.noaa.gov/psd/.

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

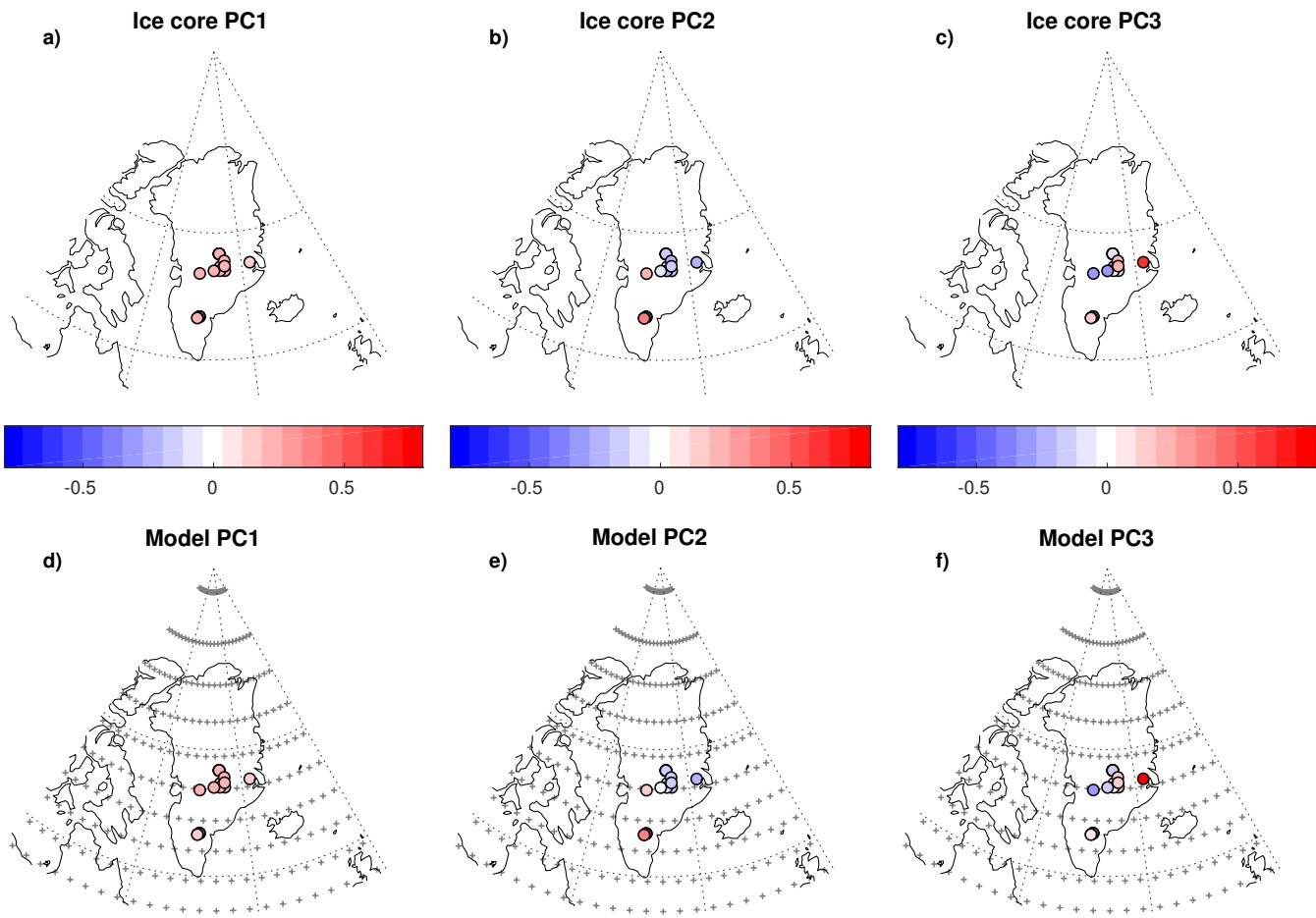

**Figure 1.** a)-c) loadings of the first three PCs of ice core $\delta^{18}$O for winter using 19 cores. d)-f) same as a)-c), but for modeled precipitation weighted $\delta^{18}$O for Nov-Apr at the sites of the 19 ice cores. Results for summer and annual data are very similar (not shown). In figure d)-f) the crosses mark the model grid showing the horizontal model resolution of $3.75^o$ x $3.75^o$.

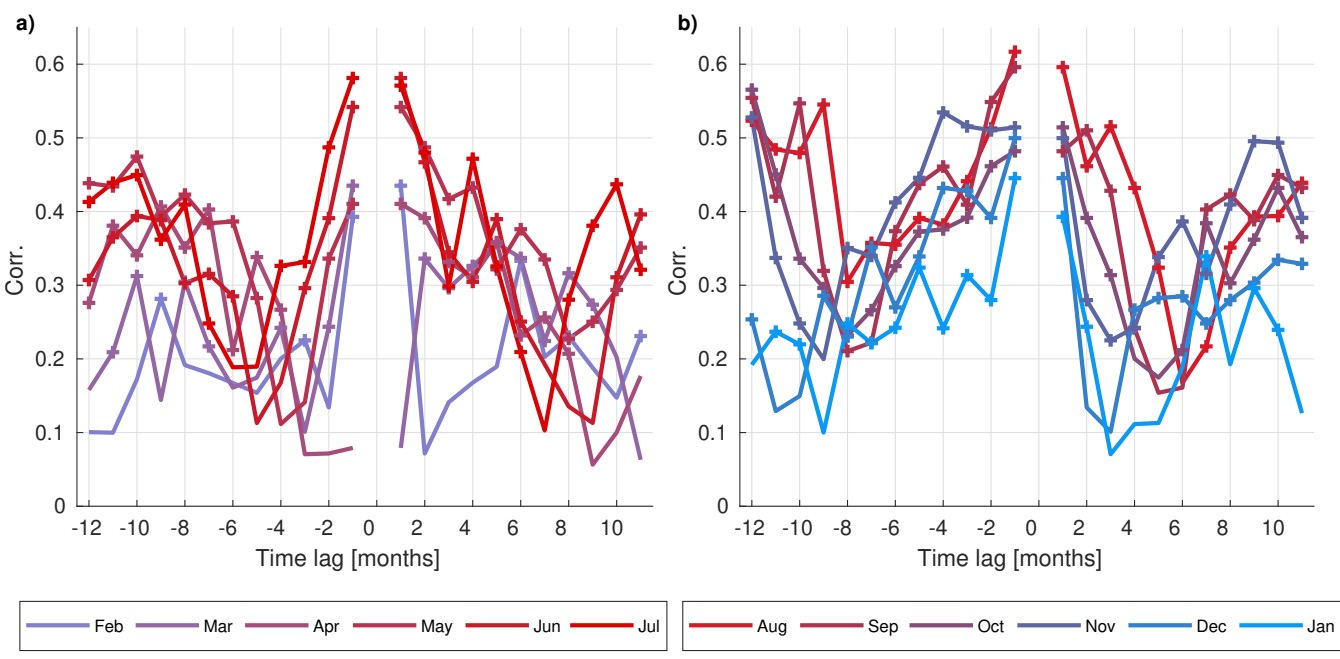

**Figure 2.** Auto-correlation analysis for PC1 of monthly 20CR SLP (1851-2010). a) shows results for Feb-Jul and b) shows Aug-Jan. (+) indicated significant correlations (p<0.01).

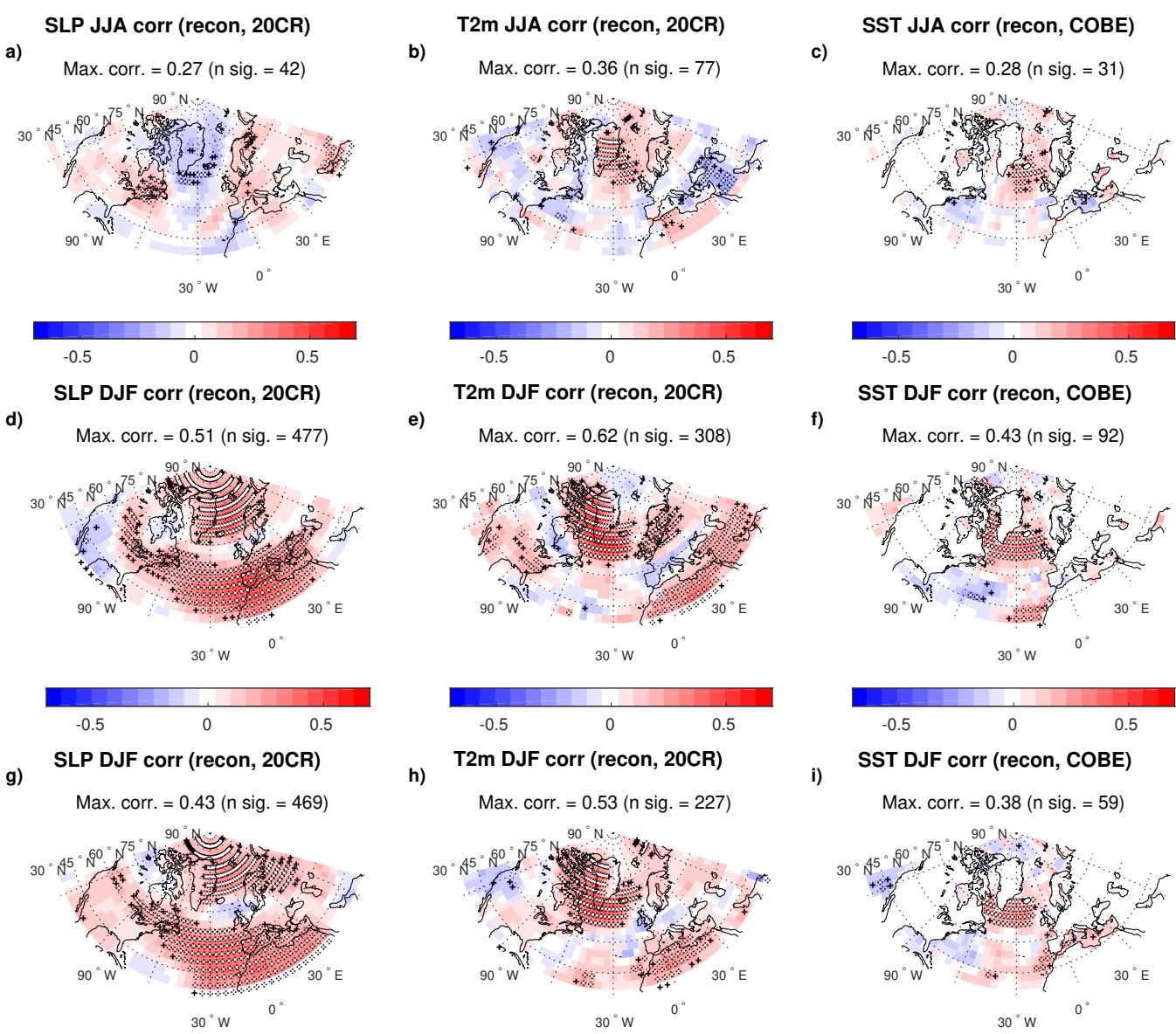

**Figure 3.** a)-c) Correlation between reconstructed (8 ice cores) and reanalysis SLP, T2m and COBE SST for JJA. The reanalysis data has been interpolated to the model grid ($3.75^o$ x $3.75^o$). Black markers indicated p<0.05 and white markers indicate p<0.025. Also indicated is the maximum correlation (Max. Corr.) and the number of significantly correlated grid points (n sig.) (p<0.05). d)-f) same as a)-c), but for DJF. g)-i) same as a)-c) but for DJF reconstructed from the winter centered annual mean ice core data. Supplementary Figure S7 shows corresponding figures for the reconstructions using 19 ice cores.

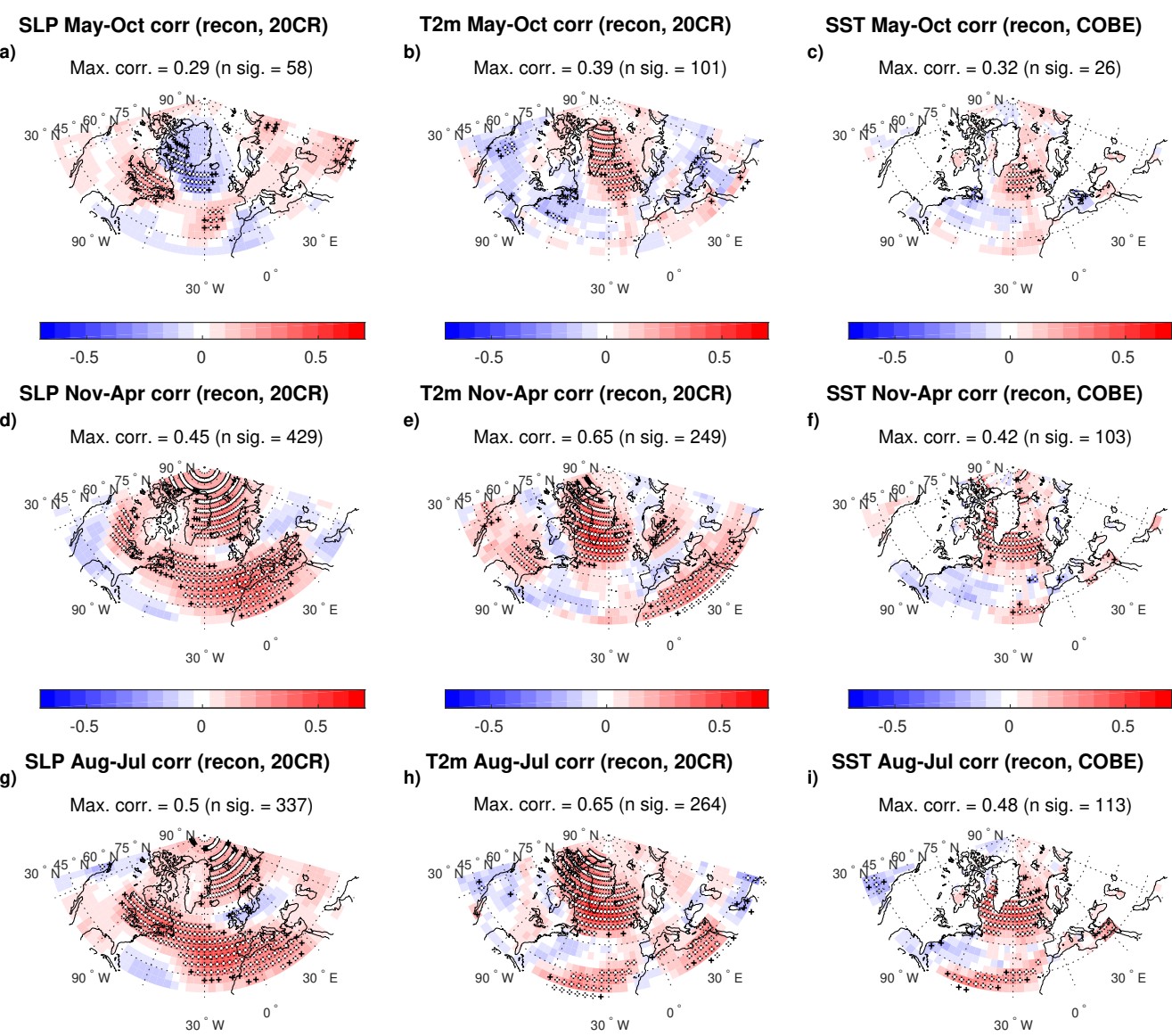

**Figure 4.** a)-c) Correlation between reconstructed (8 ice cores) and reanalysis SLP, T2m and COBE SST for sum50 (May-Oct). The reanalysis data has been interpolated to the model grid (3.75$^o$ x 3.75$^o$). Black markers indicated p<0.05 and white markers indicate p<0.025. Also indicated is the maximum correlation (Max. Corr.) and the number of significantly correlated grid points (n sig.) (p<0.05). d)-f) same as a)-c), but for Win50 (Nov-Apr). g)-i) same as a)-c), but for the winter centered annual mean (Win100, Aug-Jul). Supplementary Figure S8 shows corresponding figures for the reconstructions using 19 ice cores.

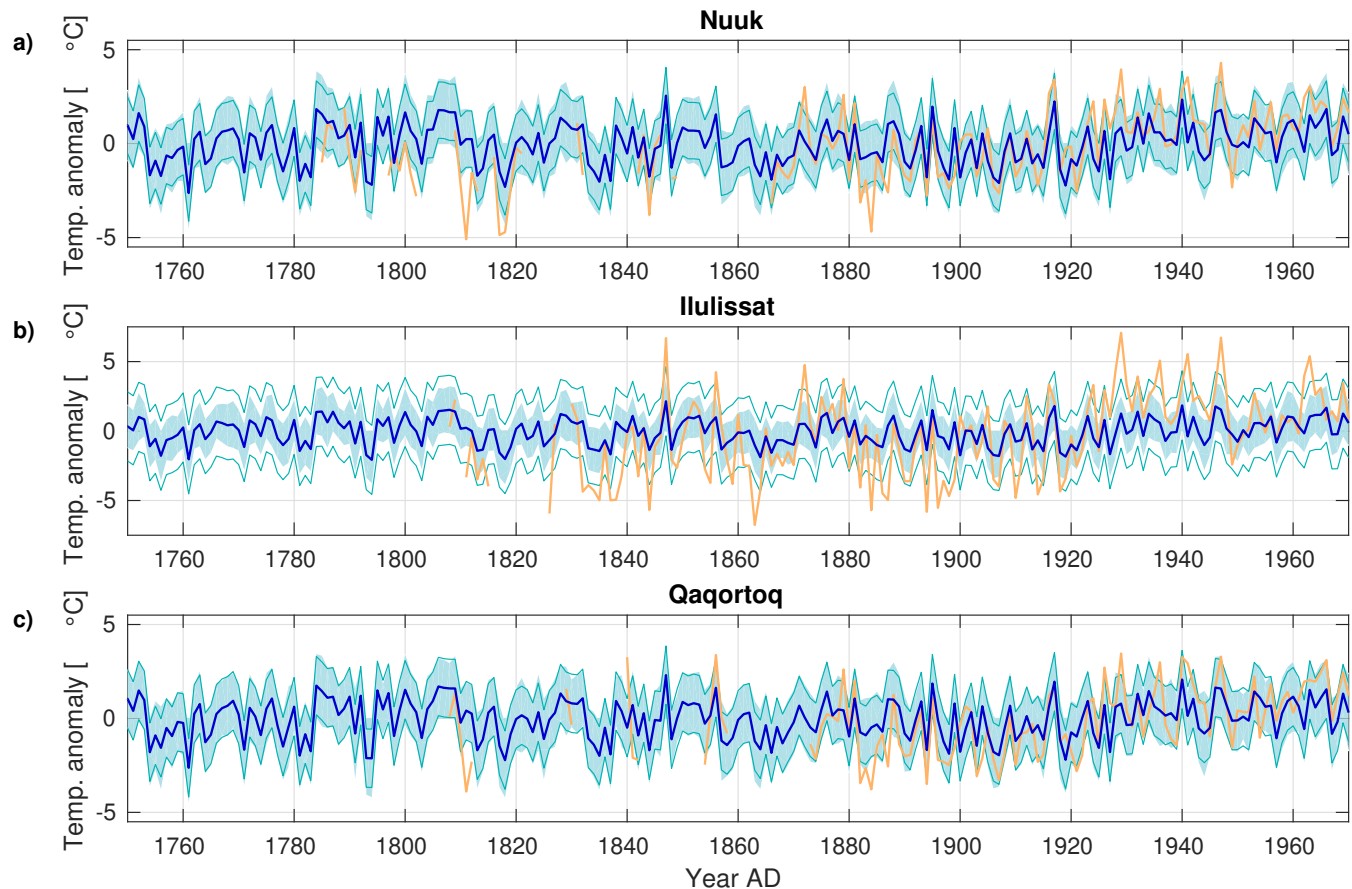

**Figure 5.** Time series of Nov-Apr (sum50) temperature for observed (yellow) and reconstructed ensemble mean (sum50, 8 ice cores) (dark blue) from Nuuk (a), Ilulissat (b) and Qaqortoq (c). Light blue shading is the one $\sigma$ spread of the reconstructed temperature and the green lines indicates the RMSE between the observed and reconstructed temperature.

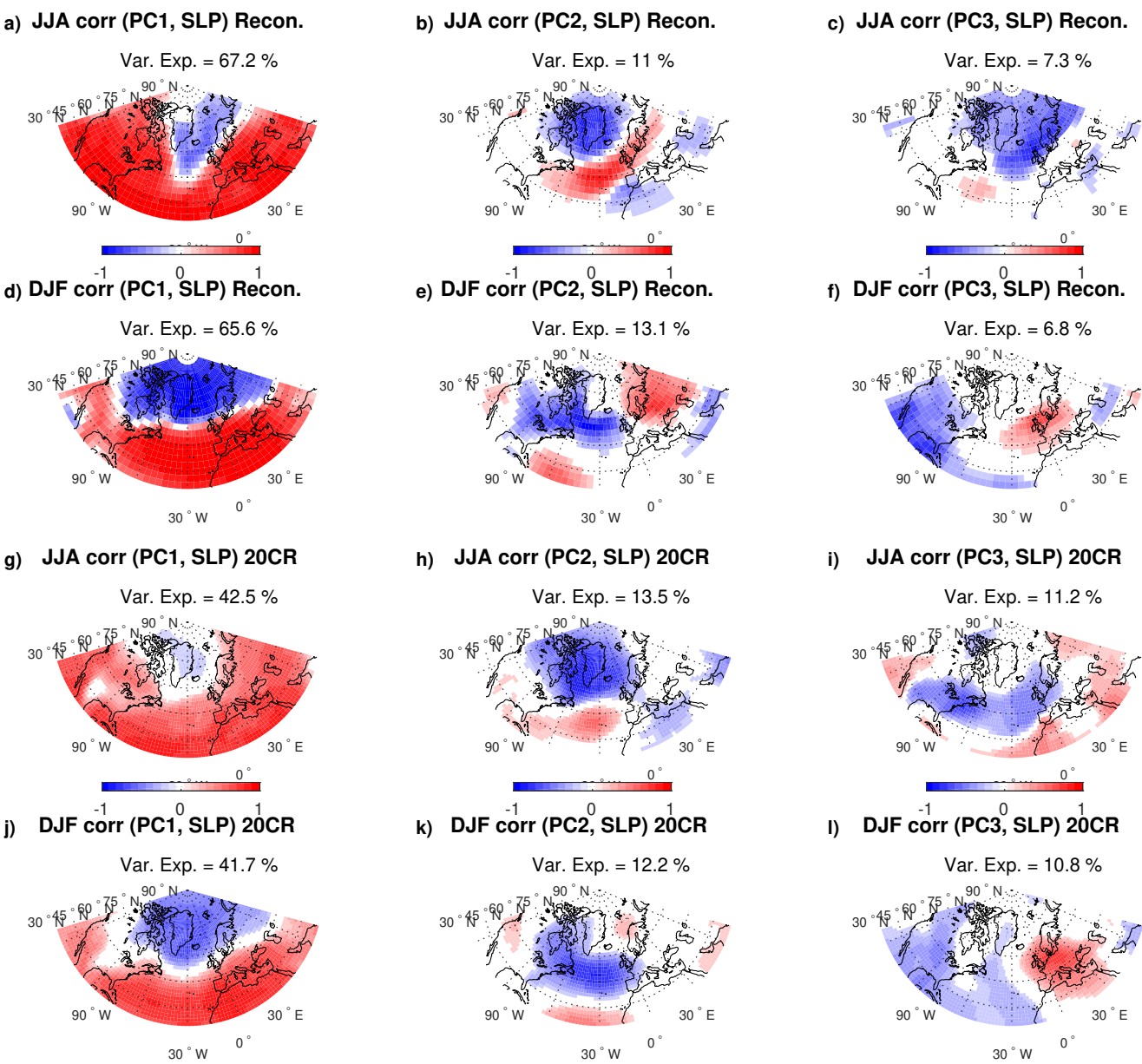

**Figure 6.** a)-c) regression of the first three reconstructed PCs of SLP on reconstructed (8 ice cores) JJA SLP, which corresponds to the reconstructed EOF patterns. d)-f) same as a)-c), but for DJF. These plots, with the addition of the plots for DJF reconstructed (8 ice cores) from the winter centered annual mean ice core data, are shown in Supplementary Figure S10, as well as corresponding plots for Sum50, Win50 and Win100 shown in Supplementary Figure S11. g)-i) regression of the first three 20CR PCs of SLP on 20CR JJA SLP, which corresponds to the EOF patterns. j)-l) same as g)-i), but for DJF. These plots for 20CR data are also shown in Supplementary Figure S12, as well as corresponding plots for Sum50, Win50 and Win100 shown in Supplementary Figure S13. The time period for all plots is 1851-1970. Only data shown for p<0.05.

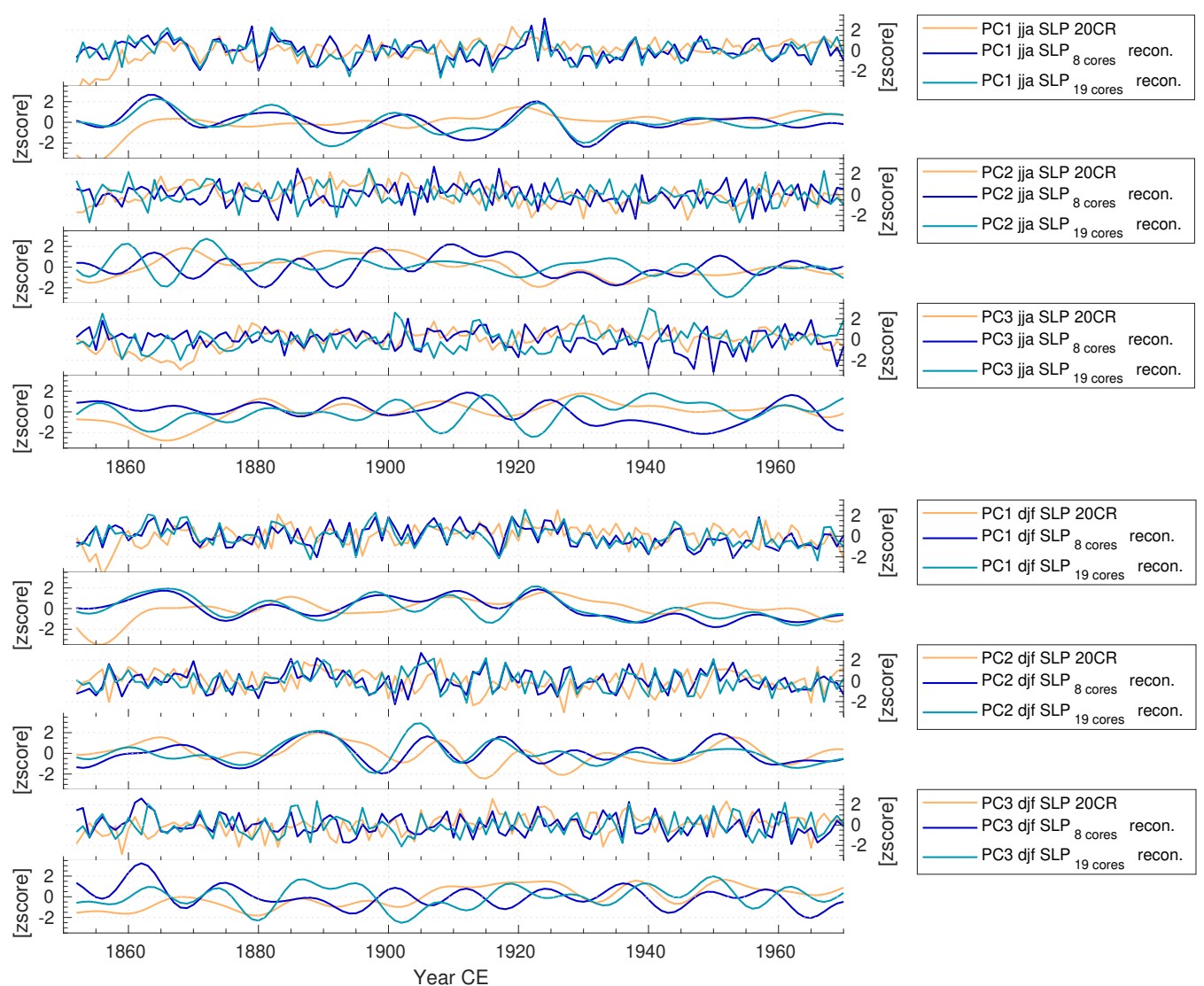

**Figure 7.** Time series of reconstructed PC1, PC2 and PC3 of SLP using 8 ice cores (dark blue) and 19 ice cores (light blue) compared to PC1, PC2 and PC3 of 20CR SLP (yellow). Smoothed curves are using a decadal FFT-filter. Top six plots are for JJA and bottom six plots are for DJF.

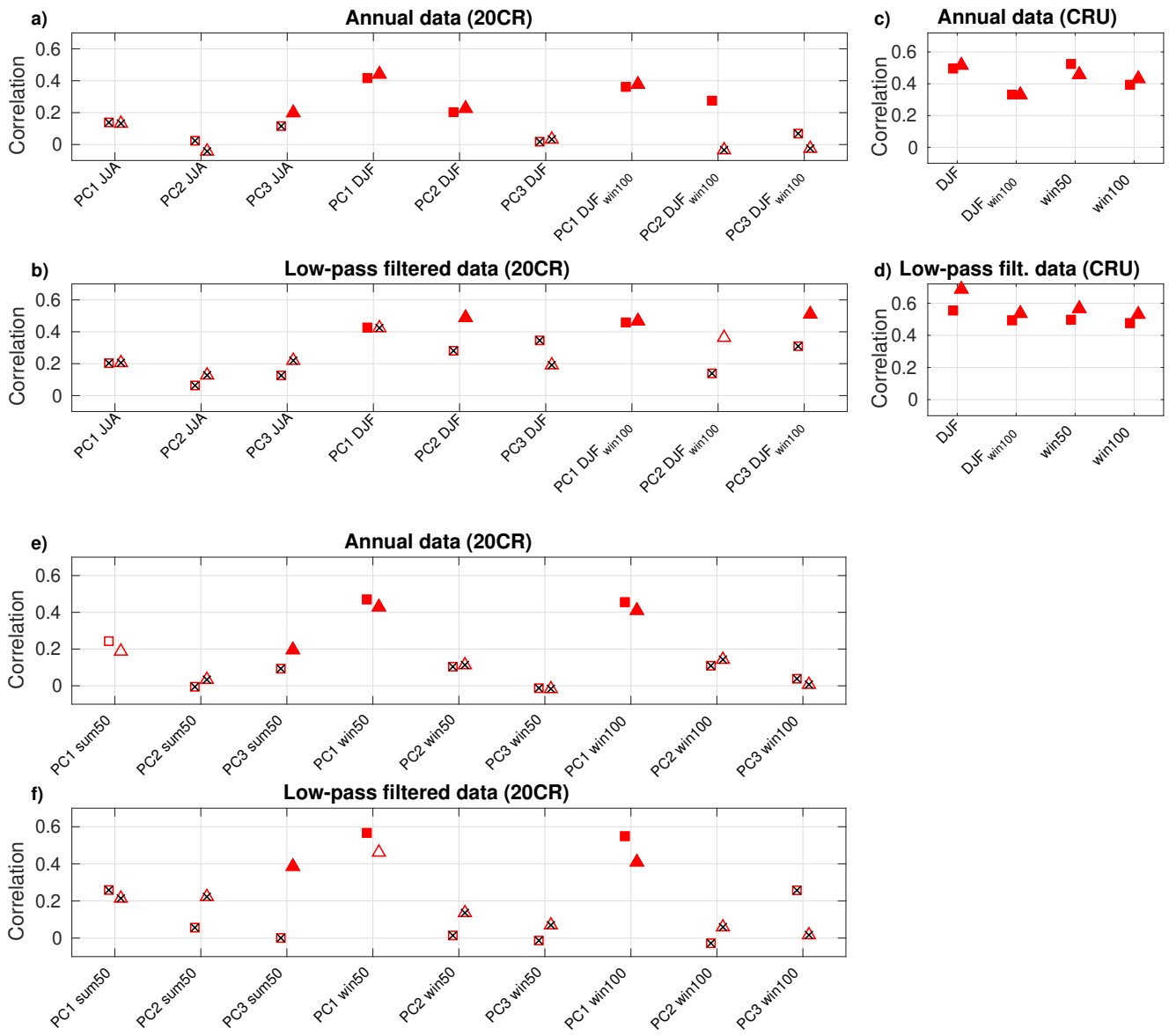

**Figure 8.** Correlation analysis for reconstructed PC1, PC2 and PC3 of SLP using 8 ice cores (triangles) and 19 ice cores (squares) correlated to PC1, PC2 and PC3 of 20CR SLP covering 1851-1970 (a)-b) and e)-f)), correlation analysis for reconstructed PC1 SLP using 8 ice cores (triangles) and 19 ice cores (squares) correlated to station based NAO covering 1824-1970 (c)-d)). The station based NAO is only valid for winter and annual data due to the seasonal shift in the centers of action. Open markers indicate significance of p<0.1 and full markers indicate p<0.05, while crossed out markers indicate p>0.1.

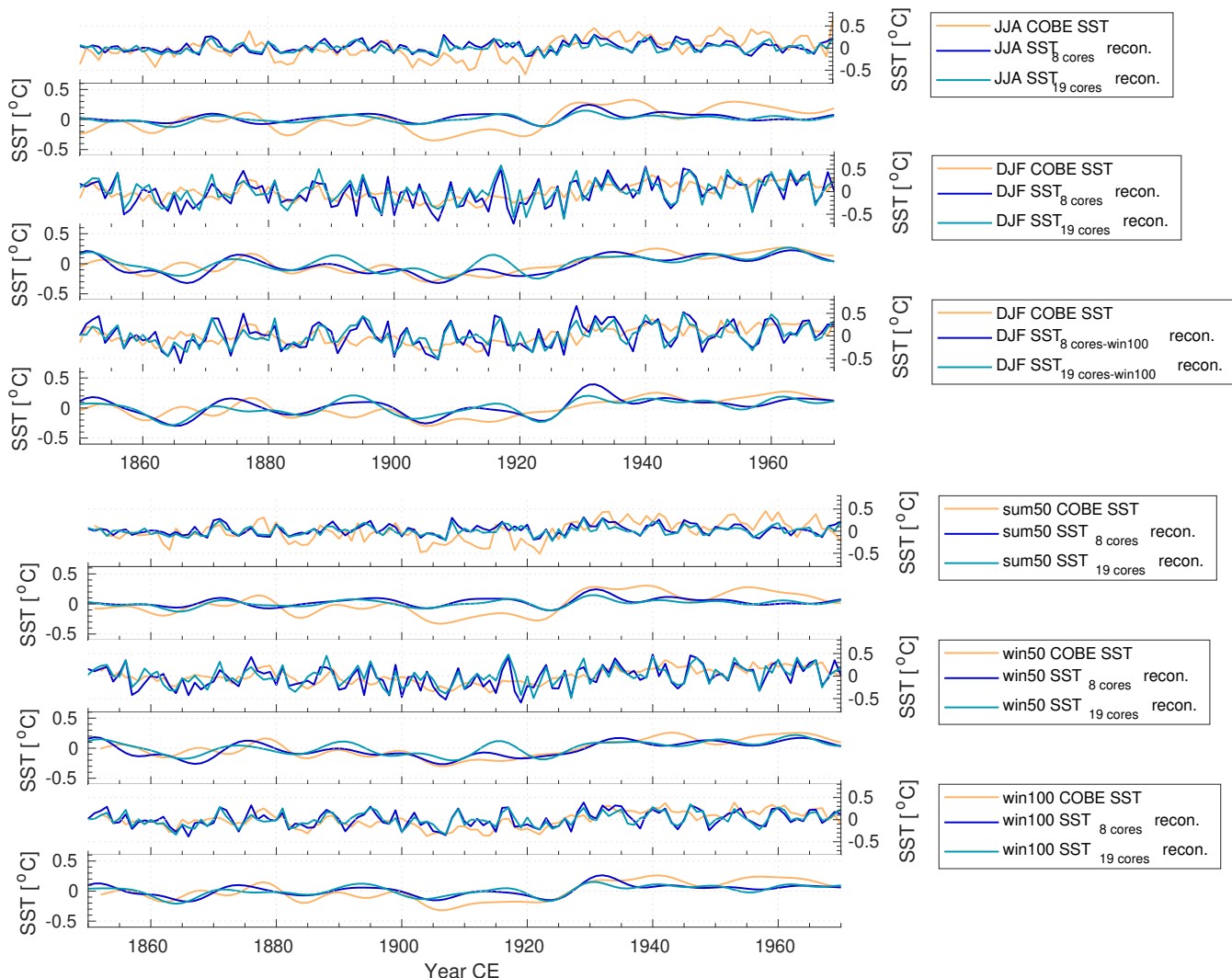

**Figure 9.** Time series of the North Atlantic SST index (50°N-70°N, 70°W-0°W) for reconstructions using 8 ice cores (dark blue) and 19 ice cores (light blue) compared to COBE SSTs (yellow). Smoothed curves are using a decadal FFT-filter. The top six plots are for JJA, DJF and DJF reconstructed using the winter centered annual mean, while the bottom six plots are for sum50 (May-Oct), win50 (Nov-Apr) and the winter centered annual mean Win100 (Aug-Jul).

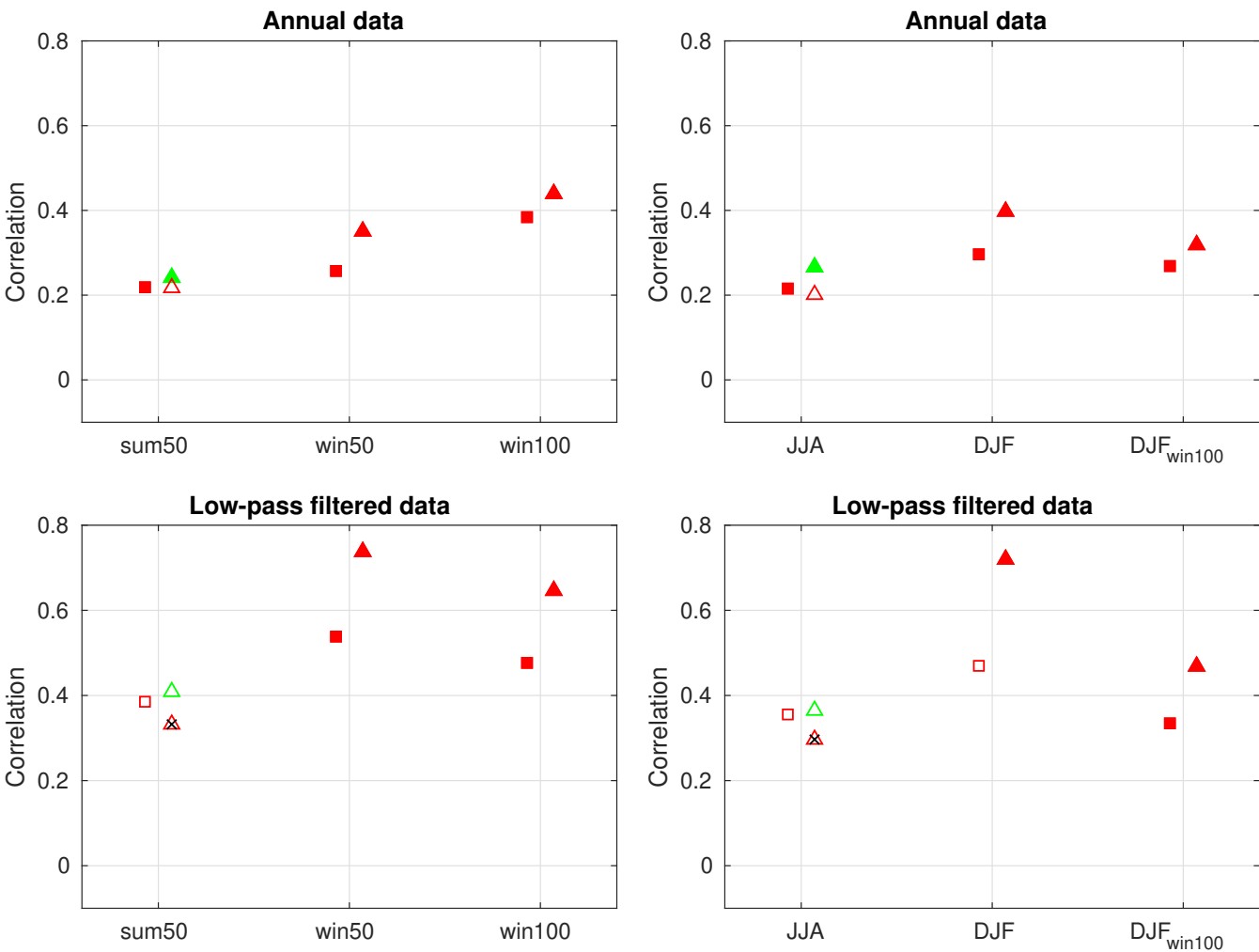

**Figure 10.** Correlation analysis of the North Atlantic SST index ($N^o$50-70$^o$N, 70$^o$W-0$^o$W) for reconstructions using 8 ice cores (triangles) and 19 ice cores (squares) correlated to COBE SSTs covering 1851-1970- The green markers are for the reconstructions including tree-ring data. Open markers indicate significance of p<0.1 and full markers indicate p<0.05, while crossed out markers indicate p>0.1.

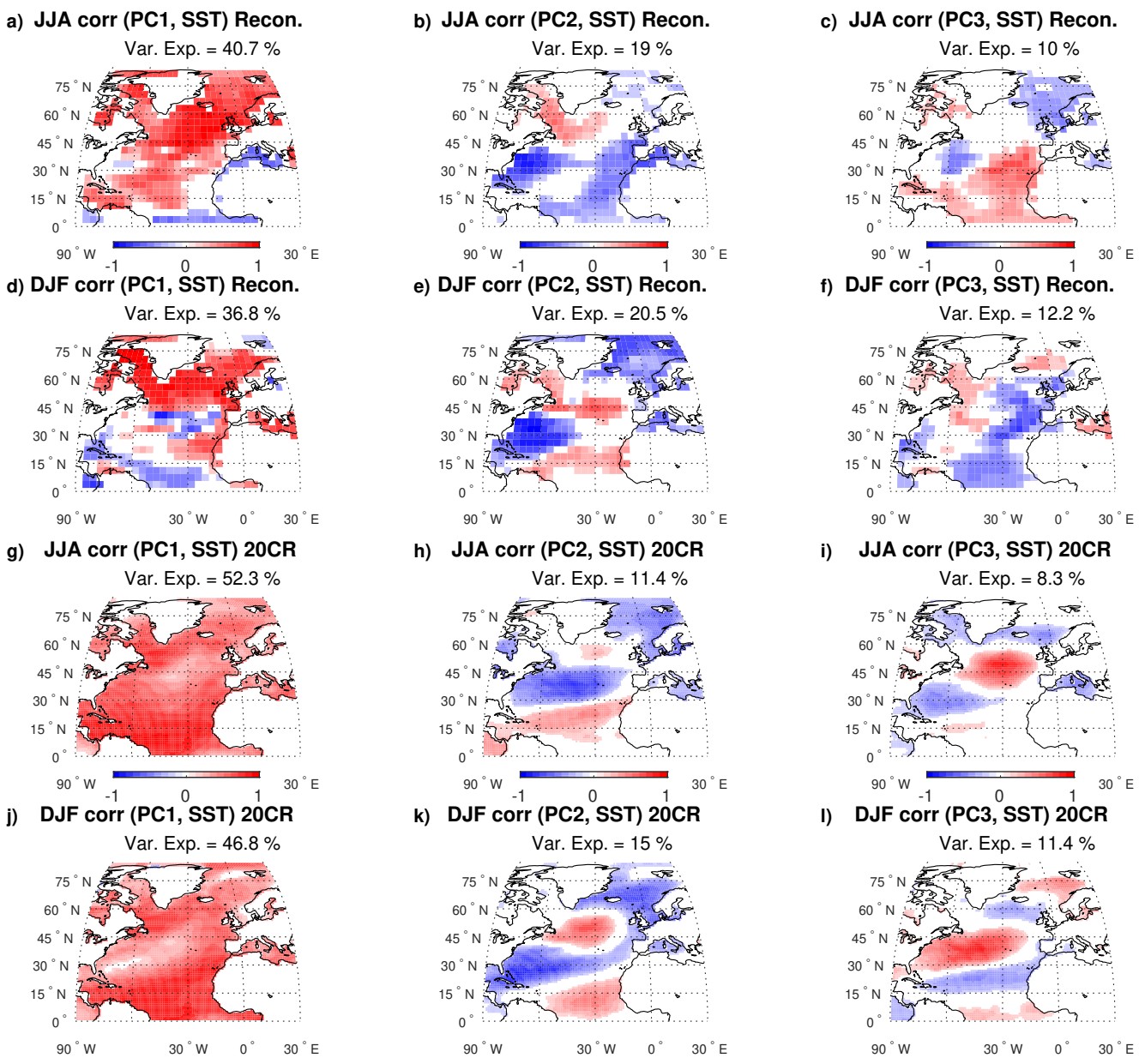

**Figure 11.** a)-c) regression of the first three reconstructed PCs of SSTs on reconstructed sum50 JJA SSTs, which corresponds to the re-constructed EOF patterns. d)-f) same as a)-c), but for DJF. Corresponding plots for reconstructions of sum50 (May-Oct), win50 (Nov-Apr) and the winter centered annual mean (Win100, Aug-Jul) are shown in Supplementary Figure S14. g)-i) regression of the first three COBE SST PCs on COBE JJA SSTs, which corresponds to the reconstructed EOF patterns. j)-l) same as g)-i), but for DJF. A corresponding figure for 20CR sum50 (May-Oct), win50 (Nov-Apr), and the winter centered annual mean Win100 (Aug-Jul) can be found in the Supplementary Figure S15. The time period for all plots is 1851-1970. Only data shown for p<0.05.

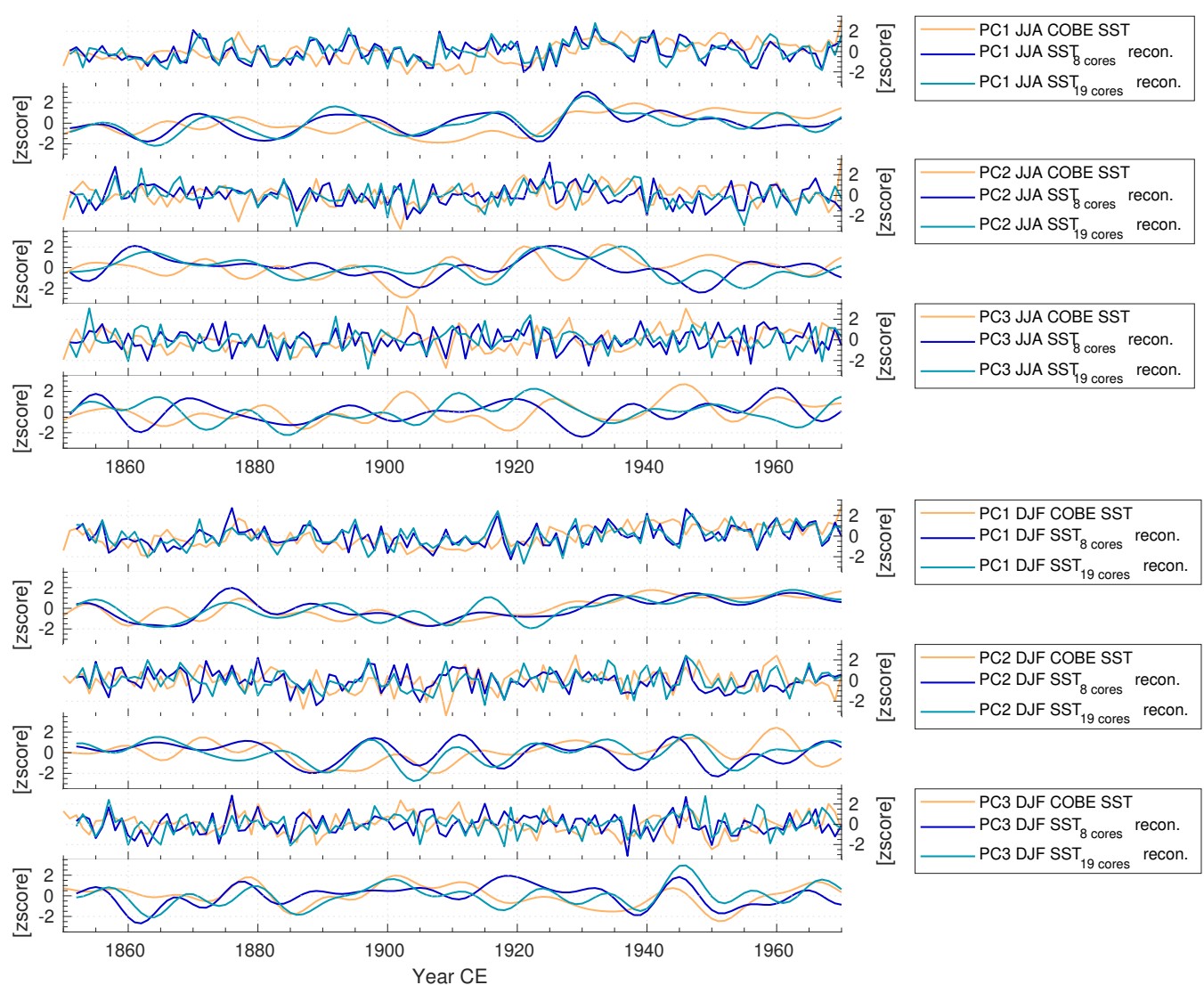

**Figure 12.** Time series of reconstructed PC1, PC2 and PC3 of SSTs using 8 ice cores (dark blue) and 19 ice cores (light blue) compared to PC1, PC2 and PC3 of COBE SSTs (yellow). Smoothed curves are using a decadal FFT-filter. Top six plots are for JJA and bottom six plots are for DJF.

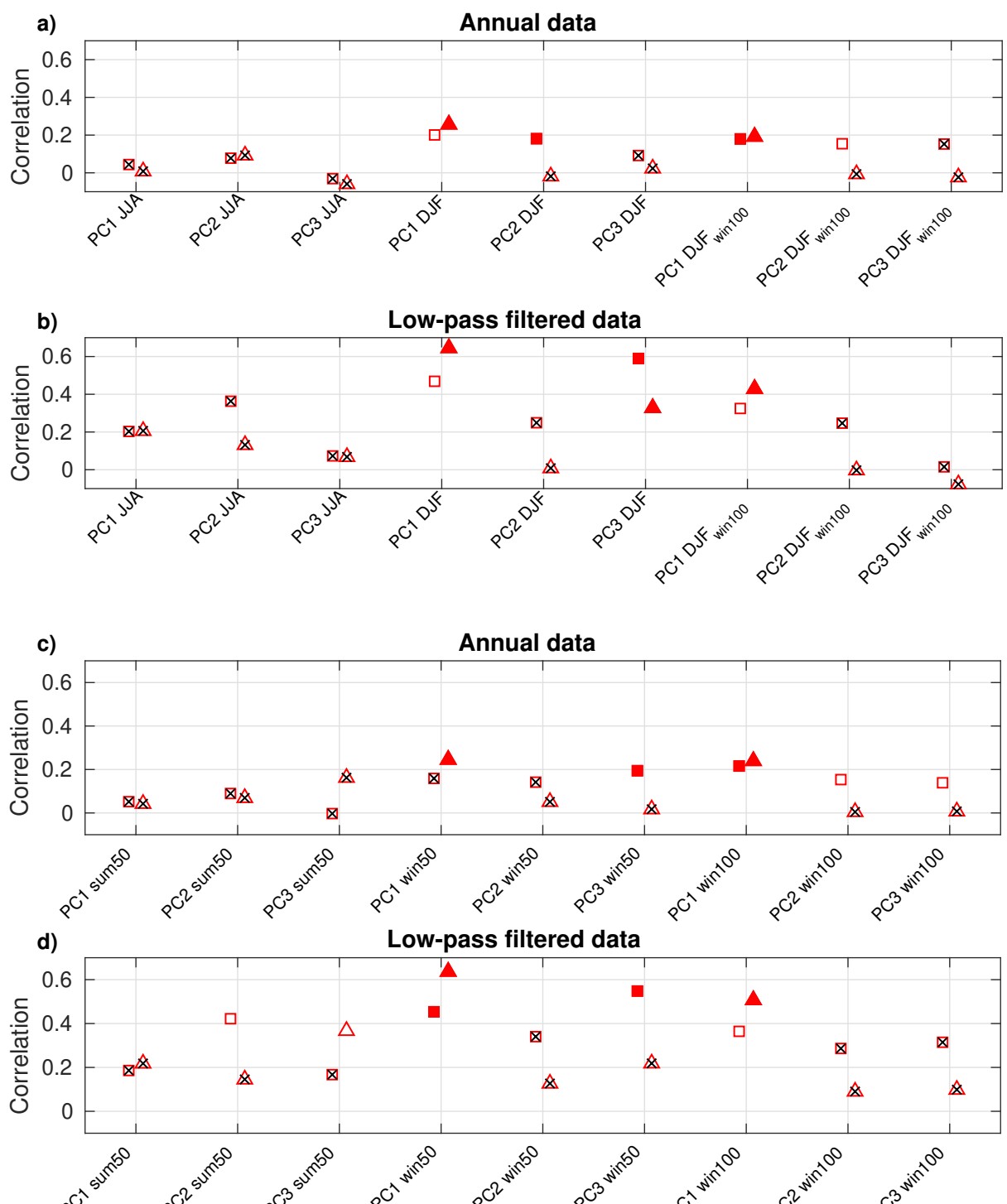

**Figure 13.** Correlation analysis for reconstructed PC1, PC2 and PC3 of SSTs using 8 ice cores (triangles) and 19 ice cores (squares) correlated to PC1, PC2 and PC3 of COBE SSTs covering 1851-1970. Open markers indicate significance of p<0.1 and full markers indicate p<0.05, while crossed out markers indicate p>0.1.

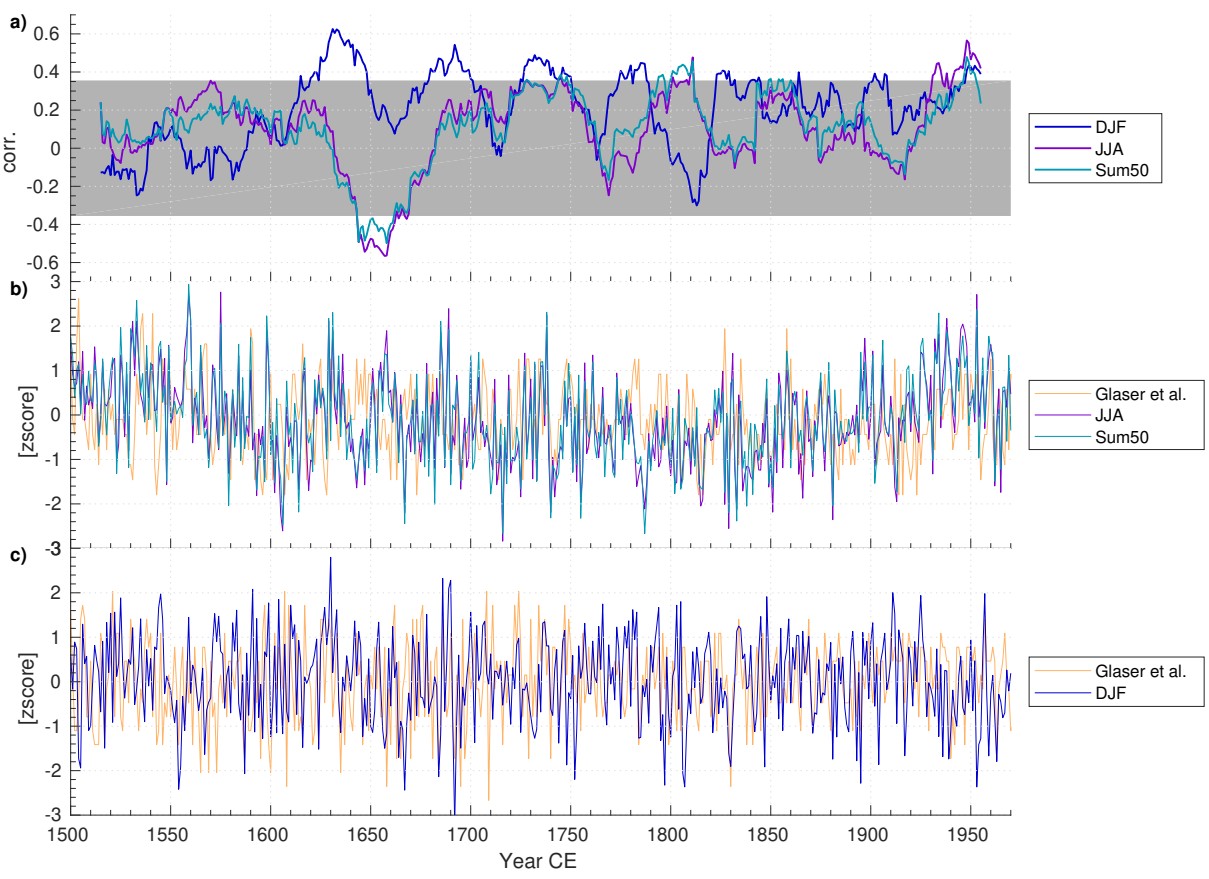

**Figure 14.** a) Moving 31-year correlation between the Glaser and Riemann (2009) Central Europe temperature index (JJA, DJF) and reconstructed temperature from this study (JJA, sum50, DJF). Correlations beyond the gray shaded area are significant (p<0.05). b) Time series of the Glaser and Riemann (2009) Central Europe temperature index (JJA) and reconstructed temperature from this study for summer (JJA, sum50). c) Time series of the Glaser and Riemann (2009) Central Europe temperature index (DJF) and reconstructed temperature from this study (DJF). For the reconstructed temperature from this study we extract the area mean temperature (T2m) for the box $50^o$N-$60^o$N and $0^o$E-$20^o$E using only values for land.

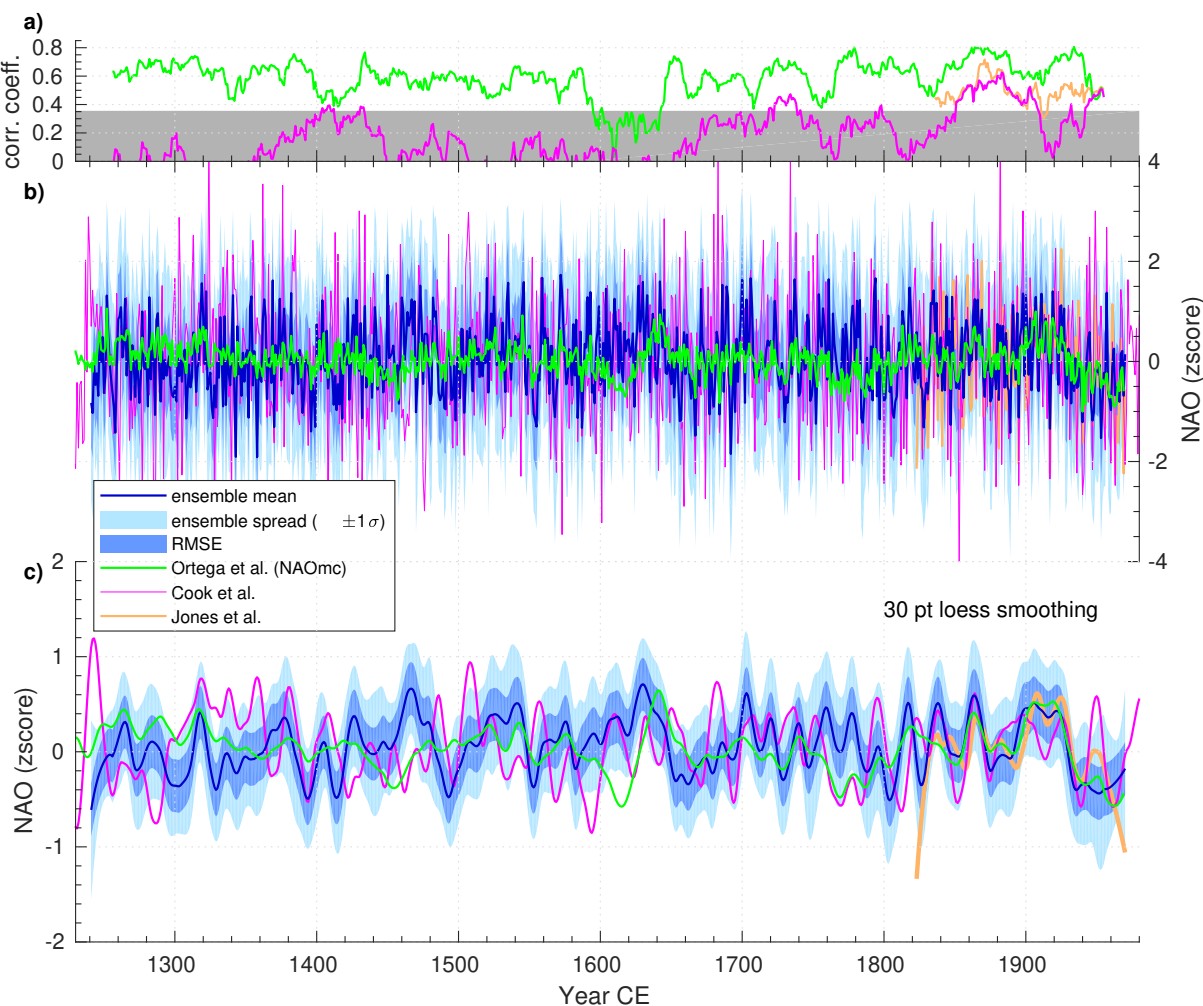

**Figure 15.** a) Moving 31-point correlation between reconstructed DJF NAO from this study and Cook et al. (2019) (magenta), Ortega et al. (2015) (green) and observed NAO (yellow) (Jones et al., 1997). Correlations beyond the gray shaded area are significant (p<0.05). b) Ensemble mean reconstructed NAO (PC1 of reconstructed SLP (Hurrell et al., 2003)) with error estimated by ensemble spread and RMSE, compared to observed NAO (Jones et al., 1997) and NAO reconstructions by Cook et al. (2019) and Ortega et al. (2015). The amplitude of all time series are scaled to fit the decadal variability of the observed NAO. c) Same as b), except filtered with a 30 point 'loess' filter.

**Table 1.** Tree-ring sites used to constrain summer reconstructions 1241-1970, with correlations to observed mean temperature from Wilson et al. (2016) for the indicated months.

| Location | Site name | Long. | Lat. | Time period | Corr. with CRUTS3.2 1901-present |
|---|---|---|---|---|---|
| Scotland | SCOT | 57.08 | -3.44 | 1200-2010 | JA: 0.75 |
| E Alps - Tyrol | TYR | 47.30 | 12.30 | 1053-2003 | JAS: 0.72 |
| Jaemtland | JAEM | 63.30 | 13.25 | 783-2011 | AMJJAS: 0.75 |
| Tjeggelvas, Arjeplog, Ammarnäs composite | TAA | 65.54-66.36 | 16.06-18.12 | 1200-2010 | MJJA: 0.81 |
| North Fenno | EFmean | 66-69 | 19-32 | 750-2010 | JJA: 0.76 |
| Forfjorddalen | FORF | 68.47 | 15.43 | 978-2005 | JA: 0.71 |
| Tatra | TAT | 48-49 | 19-20 | 1040-2010 | MJ: 0.45 |
| South Finland | SFIN | 62.19.30 | 28.19.30 | 760-2000 | MJJA: 0.71 |

**Table 2.** Reconstructions featured in this study. A total of twelve reconstructions are done using 6 data sets, e.g. both the reconstructions for JJA and Sum50 use the same ice core data representing the summer season May-Oct, but targeting the differently defined summer seasons by extracting either JJA or May-Oct from the model output. The number of ensemble members (no. ens.) are given in parenthesis for each set of seasons. The winter reconstruction for DJF using 8 ice cores covering 1241-1970 is published in Sjolte et al. (2018).

| Data set and time span | 19 cores, 1777-1970 | 8 cores, 1241-1970 |
|---|---|---|
| Seasons (no. ens.) | JJA/Sum50 (31) | JJA/Sum50 (39) |
| Seasons (no. ens.) | DJF/Win50 (34) | DJF/Win50 (39) |
| Seasons (no. ens.) | DJF/Win100 (33) | DJF/Win100 (39) |

**Table 3.** Summary of maximum correlations (Max. corr.) and number of grid points with significant correlation (n sig., p<0.05) from Figure 3 and 4, as well as Supplementary Figure S7, S8 and S9. Results in columns Sum50* and JJA* are for the reconstructions using tree-ring data.

| | JJA Short | JJA Long | JJA* Long | Sum50 Short | Sum50 Long | Sum50* Long | DJF Short | DJF Long | $DJF_{win100}$ Short | $DJF_{win100}$ Long | Win50 Short | Win50 Long | Win100 Short | Win100 Long |
|---|---|---|---|---|---|---|---|---|---|---|---|---|---|---|
| $Corr_{SLP}$ | 0.32 | 0.27 | 0.33 | 0.31 | 0.29 | 0.26 | 0.48 | 0.51 | 0.39 | 0.43 | 0.48 | 0.45 | 0.43 | 0.50 |
| $n_{SLP}$ | 86 | 42 | 120 | 69 | 58 | 61 | 433 | 477 | 511 | 469 | 451 | 429 | 310 | 337 |
| $Corr_{T2m}$ | 0.40 | 0.36 | 0.63 | 0.38 | 0.39 | 0.59 | 0.63 | 0.62 | 0.56 | 0.53 | 0.62 | 0.65 | 0.62 | 0.65 |
| $n_{T2m}$ | 116 | 77 | 247 | 135 | 101 | 266 | 283 | 308 | 218 | 227 | 311 | 249 | 225 | 264 |
| $Corr_{SST}$ | 0.31 | 0.28 | 0.50 | 0.36 | 0.32 | 0.49 | 0.41 | 0.43 | 0.42 | 0.38 | 0.43 | 0.42 | 0.48 | 0.48 |
| $n_{SST}$ | 71 | 31 | 116 | 56 | 26 | 138 | 85 | 92 | 95 | 59 | 95 | 103 | 124 | 113 |

**Table 4.** Correlation between reconstructed and observed temperature for Greenland coastal stations (1874-1970) and the Icelandic station, Stykkisholmur (1831-1970). Bold marks p<0.05, (*) marks p<0.10. The low pass filter is a decadal FFT filter.

| 19 ice cores | Sum50 | Low pass | Win50 | Low pass | Win100 | Low pass |
|---|---|---|---|---|---|---|
| Stykkisholmur | **0.32** | **0.48** | **0.33** | **0.44** | **0.25** | 0.17 |
| Nuuk | **0.19** | 0.33 | **0.58** | 0.35 | **0.52** | **0.47** |
| Ilulissat | 0.18* | 0.35 | **0.53** | **0.48** | **0.45** | **0.40** |
| Qaqortoq | **0.24** | 0.34 | **0.56** | 0.40* | **0.53** | 0.44* |
| SWG index | **0.22** | 0.38* | **0.59** | 0.42* | **0.52** | **0.44** |

| 8 ice cores | Sum50 | Low pass | Win50 | Low pass | Win100 | Low pass |
|---|---|---|---|---|---|---|
| Stykkisholmur | **0.33** | 0.38* | **0.33** | **0.53** | **0.28** | 0.37* |
| Nuuk | **0.24** | 0.36* | **0.60** | **0.53** | **0.58** | **0.56** |
| Ilulissat | **0.19** | 0.39* | **0.56** | **0.60** | **0.50** | **0.52** |
| Qaqortoq | **0.27** | **0.45** | **0.60** | **0.65** | **0.59** | **0.59** |
| SWG index | **0.26** | **0.44** | **0.63** | **0.63** | **0.58** | **0.56** |

| 19 ice cores | JJA | Low pass | DJF | Low pass | $DJF_{win100}$ | Low pass |
|---|---|---|---|---|---|---|
| Stykkisholmur | **0.27** | **0.58** | **0.35** | **0.41** | **0.27** | 0.20 |
| Nuuk | 0.10 | -0.03 | **0.56** | **0.49** | **0.45** | **0.45** |
| Ilulissat | 0.09 | 0.27 | **0.56** | **0.67** | **0.41** | **0.42** |
| Qaqortoq | **0.22** | 0.43* | **0.52** | **0.49** | **0.48** | 0.41* |
| SWG index | 0.15 | 0.29 | **0.58** | **0.57** | **0.47** | **0.45** |

| 8 ice cores | JJA | Low pass | DJF | Low pass | $DJF_{win100}$ | Low pass |
|---|---|---|---|---|---|---|
| Stykkisholmur | **0.29** | **0.50** | **0.27** | **0.41** | **0.29** | 0.34 |
| Nuuk | 0.18* | 0.14 | **0.57** | **0.53** | **0.49** | **0.47** |
| Ilulissat | 0.07 | 0.29 | **0.58** | **0.70** | **0.47** | **0.55** |
| Qaqortoq | **0.25** | 0.43* | **0.57** | **0.62** | **0.52** | **0.50** |
| SWG index | **0.21** | 0.35 | **0.61** | **0.67** | **0.52** | **0.52** |