# Peer review of "Figure S1: Locations of sites for the 8 ice cores for the period 1241-1970 (black) and the 19 cores covering 1778-1970 (blue). The 19 cores include a total of 5 cores from DYE-3 and 6 cores from GRIP, as well as Site A, B, D, E and G located close to Crete."

_Climate of the Past, 2019_

## Referee Comment (RC1) · Anonymous Referee #1 · 25 Dec 2019

The manuscript presents several reconstructions of climate fields in the North Atlantic over the last centuries. The reconstructions are based on proxy data on the one hand (oxygen isotope concentrations in Greenland ice cores and on European dendroclimatological data ) and on climate simulations with a isotope enabled climate model. Both types of data are combined applying the analog method. The manuscript is mainly focused on testing the sensitivity of the reconstructions on the number of ice cores, on the definitions of the target seasons. The manuscript is less focused on the physical results and the implications of these reconstructions. It is therefore rather technical

manuscript. There are almost no results or dicussion on the physical mechanisms or past climate variations. This is in principle fine, but the readers should be made aware of this early on in the abstract and in the introduction. In my opinion, the manuscript addresses interesting issues. For instance, the use isotopic data from ice cores in combination with climate simulations to reconstruct the atmospheric circulation is an interesting idea. However, being this a more technical manuscript, the description of the methods applied leaves many open questions for the reader. This description should be considerably improved. At some points it is so unclear that I had doubts about the correct application of the method, although I hope that it is in the end correct.

Main points:

1. One main concern is the limited methodological description. The authors apply the analog method after a pre-filtering by Principal Component Analysis, but it is not cleat how this pre-filtering is actually conducted. Important questions that may impact the results :

1.1 Are the PCs calculated from the covariance of correlation matrix?

1.2 in Equation 1, are the PCS normalized to unit variance or has each PC the platitude representing the corresponding explained variance. This is important because this point represents two options: all PCs are equally weighted for the calculation of the distance of the analogues, or each PCs is weighted according to the varianve it represents. The selected analogues are different depending on the option chosen.

1.3 More importantly, for a correct application of the method, the EOFs patterns (derived from ice-core records and from model grid cells) associated to each PC in equation 1 must be the same . Otherwise, the PC-coordinates PC_{model} and PC_{icecores} ) would not be linked to the same EOFs and therefore would represent coordinates in different subspaces. The calculated distances would not be meaningful. It may happen that the patterns of EOF_{model} and EOF_{icecores} are very similar, in which case this problem would be minor. But this is not guaranteed. A way to ensure that the PCs in equation 1 refer to the same EOF patterns is to use, for instance, EOF_{model} for both data sets and calculate PC_{icecores} by projecting the ice core anomalies onto EOF_{model} . (Or viceversa, use EOF_{icecores} for both data sets).

This is the point that most strongly worries me. If EOF_{model} and EOF_{icecore} patterns are really different the whole application of the method is not correct, and all results should be re-calculated.

2. Another unclear methodological point is how the dendro data are included for the reconstruction. Here, I cannot make any useful suggestion because the authors approach remains unclear to me. This needs to be much better explained:

2.1 Are the analogues searched using ice-core and dendro data simultaneously, i.e. a 12-month-long model analog have to be close to the icecore data in the target season and close to the dendro-reconstructed summer temperatures in the summer season?. If yes, how is the EOF filtering implemented here? How many 'temperature' EOFs are used.

or

2.2 Do the selected analogues (using ice core data) undergo a secondary selection procedure targeting the dendro-reconstructed temperatures ?

2.3 In both cases, are the distances to the dendro-data and the distances to the ice core data equally weighted ? How is this implemented if the number of EOFs for each type of data sets is presumably not the same.

3. The validation of the results is essentially made by calculating the correlation between reconstructions and 20CR reanalysis. However, the amplitude of the reconstructions is not validated. This may be important because the amplitude of reconstructed variability may depend on the number of analogs selected: best-analog-selection (just one analogue) will roughly produce the same amplitude of variations, although the validation correlation will be lower; in contrast, using the mean of a larger number of

analogues subdues the variability, and this effect can be substantial when using 39 (?) analogues. There is an unavoidable trade-off between better correlations and more realistic amplitudes, as shown in Gomez-Navarro et al. Pseudo-proxy tests of the analogue method to reconstruct spatially resolved global temperature during the Common Era, Clim. Past, 13, 629–648, https://doi.org/10.5194/cp-13-629-2017, 2017 ).

Particular points:

4. We test a range of climate reconstructs varying the definition of the seasons

climate reconstructions

5. The abstract does not mention the reconstruction method at all, despite the manuscript being essentially methodological in nature.

6. best captured when defining the season December-February

the season as December-February

7. line 10 best captured when defining the season December-February due to the dominance of large scale patterns, while for summer the weaker, albeit more strongly auto-correlated, variability is better captured using a longer season of May-

This sentence becomes clearer later in the manuscript. Here, I would suggest to improve its clarity, for instance, using 'more persistent in time' instead of autocorrelated.

8. One point that sets the study by Sjolte et al. (2018) apart from the other studies mentioned in this section, is the use of

delete comma after section

9. line 145 For the summer reconstructions also using tree-ring data we sort the 39 existing ensemble members

I am rather confused by this sentence. The number 39 is mentioned here for the first time, if I am not mistaken. What are these ensemble members? are they the analogs

previously selected targeting the ice-core data ?

10. line 155 In this study we follow the convention of using the term PCs for the time series of the main modes of variability, while using the term EOFs for the spatial patterns of the modes. The method of Ebisuzaki (1997) is used to calculate the significance when

this sentence should appear before equation 1, at the very least

11. line 163 A key factor in how well seasonal climate reconstructions can represent climate itself, is the auto-correlation structure of atmospheric

climate itself ? I guess the authors mean to what extent can seasonal proxy data represent annual means ?

12. line 167 Figure 2 shows the monthly auto-correlation of each month of the PC-based NAO calculated from the 20CR. These figures show that during the cold season the

Which is the PC that represents the NAO ? Here, it is assumed that, for each month the leading PC is the NAO. In summer this is not always the case, and it depends on the geographical region selected to conduct the PCA.

13. line 180

circulation modes. We do this by performing monthly reconstructions for pressure and evaluating the resulting main modes of circulation against the modes of the 20CR. This is done using the same method as for the seasonal reconstructions,

what does 'evaluating the modes' mean here ?. The spatial patterns (EOFs) resulting from an analog reconstructions can be very realistic irrespective of the skill of the analog method. The analog method is just a resampling from a data pool. A random resampling of SLP from the 20CR reanalysis or from a model run will produce the same EOFs as the orginal data set, so even if the analog is wrongly implemented, the resulting EOFs may look correct. This is different for a temporal validation, e.g. correlation between reconstructions and observations, where the skill of the analog selection is critical. The authors should be here more specific.
* * *

---

## Referee Comment (RC2) · Anonymous Referee #2 · 11 Jan 2020

The manuscript by Sjolte et al. investigates a new innovation in the rapidly developing field of paleoclimate data assimilation. Specifically, they investigate the potential of reconstructing seasonal fields using subannually resolved ice-core (and to a lesser extent, tree ring) data from the North Atlantic. The manuscript is well-written, well-illustrated and generally well-organized, and the results are interesting, and suited to Climate of the Past. I do however, have a few concerns and and suggested additions to the manuscript that I'd like to see addressed.

**Major Issues**

[Figure]

In general, in my opinion, the primary weakness of the manuscript is that the exploration of the reanalyses is rather limited. For example, in the authors subdivide the ice cores into a group of 8 that extends from 1241-1970 and a larger group that is shorter (1777-1970). However the reconstructions are only analyzed in the context of instrumental data. No results from prior to 1850 are shown in the manuscript or supplement, except for figure S3, which is specifically focused on the tree ring sites. In evaluating this technical approach, it is important for readers to be able to see how the longer term variability compares to other reconstructions from the region, and to consider and discuss how the seasonal assimilations affect long-term variability, and the potential climatic implications of that. Given that this approach creates a field reconstruction; these results could be compared to regional temperature reconstructions, NAO reconstructions, and more, and give the readers a better sense of how this approach compares with previous efforts.

At present, the evaluation of the results is restricted to spatial comparison of the first three PCs with instrumental data, temporal comparison of the same thing. I was glad to see SSTs averaged and compared to instrumental data, but feel like the comparison was ultimately very limited.

The other major weakness of the manuscript, that I believe should be able to address, was the representation of uncertainty. The methodological approach to uncertainty quantification; an ensemble based approach, is reasonable. I was disappointed however that the results were not presented in the manuscript. Every figure in the manuscript, except for the first two, could, and should, have uncertainty ranges (like 95

**Additional issues/notes**

I'm a little confused about how the analog matching is working, based on figure 1. Specifically, are any adjustments made to the model-output before calculating the EOFs of $\delta^{18}$O? If not, I'm confused about how there is such fine spatial structure in the model $\delta^{18}$O, given that it has 3.5 degree grid cells. In fact, I think it would be helpful

to see the outlines of the gridcells on the lower half of Figure 1. Maybe there's enough resolution there, but I found it confusing. I'm also pretty surprised about how comparable the modeled and observed $\delta^{18}$O EOFS are, they're nearly identical. I'm not particularly familiar with this region and proxy, but model-proxy EOF comparisons this similar are exceptionally rare, unless one was forced/derived from the other, and I'd be interested to learn more about this.

Here's a suggestion that might be beyond the scope of this manuscript, but that I think is interesting: have you considered trying to assimilate different proxies for different seasons, but for the same assimilation? It would be really interesting to see what an annual reconstruction looks like where tree rings were assimilated for summer, while ice cores were simultaneously assimilated for winter – i.e., do the analog matching differently for each season but find the years that match both optimally.

**Minor issues:**

Line 7. "Reconstructs" should be "Reconstructions" 32: 18-O should have the standard superscript formatting 85: "extend" should be "extent" 328: "depended" should be "dependent" 385: "particularly" should be "particular"

Figure 7: Add some additional labels to the panels to help differentiate. It took me awhile to figure out why c and d were separated.

---

## Author Comment (AC2) · 11 Feb 2020

The manuscript by Sjolte et al. investigates a new innovation in the rapidly developing field of paleoclimate data assimilation. Specifically, they investigate the potential of reconstructing seasonal fields using subannually resolved ice-core (and to a lesser extent, tree ring) data from the North Atlantic. The manuscript is well-written, well-illustrated and generally well-organized, and the results are interesting, and suited to Climate of the Past. I do however, have a few concerns and and suggested additions to the manuscript that I'd like to see addressed.

- *We thank the reviewer for the positive comments and interest in out manuscript, as well as the detailed comments which helped us greatly improve the manuscript.*

Major Issues

In general, in my opinion, the primary weakness of the manuscript is that the exploration of the reanalyses is rather limited. For example, in the authors subdivide the ice cores into a group of 8 that extends from 1241-1970 and a larger group that is shorter (1777-1970). However the reconstructions are only analyzed in the context of instrumental data. No results from prior to 1850 are shown in the manuscript or supplement, except for figure S3, which is specifically focused on the tree ring sites. In evaluating this technical approach, it is important for readers to be able to see how the longer term variability compares to other reconstructions from the region, and to consider and discuss how the seasonal assimilations affect long-term variability, and the potential climatic implications of that. Given that this approach creates a field reconstruction; these results could be compared to regional temperature reconstructions, NAO reconstructions, and more, and give the readers a better sense of how this approach compares with previous efforts.

- *We agree that an in-depth comparison to other reconstructions of the longer term variability would be very interesting. For the DJF NAO reconstruction covering 1241-1970 this is already done by Sjolte et al. (2018). The purpose of this paper is to test detailed aspects of seasonal variability and which factors affect the skill of seasonal reconstructions. These tests are only possible with observation-based data where we have full control on the time scale and seasonality. Furthermore, a full-scale comparison to the long term variability of other reconstructions is a whole study in itself, and this manuscript is already crowded by many figures and results. As also pointed out by Referee #1 the scope of this study should be better defined. We will do so in the revised manuscript.*

At present, the evaluation of the results is restricted to spatial comparison of the first three PCs with instrumental data, temporal comparison of the same thing. I was glad to see SSTs averaged and compared to instrumental data, but feel like the comparison was ultimately very limited.

- *Please note that we also compare to the station-based NAO (Figure 7c and 7d), as well as temperature data from Greenland and Iceland (Table 4). Again, the motivation for using only observation-based data is that this is the only data where there is no uncertainly with respect to sub-annual temporal resolution. See also reply above.*

The other major weakness of the manuscript, that I believe should be able to address, was the representation of uncertainty. The methodological approach to uncertainty quantification; an ensemble based approach, is reasonable. I was disappointed however that the results were not presented in the manuscript. Every figure in the manuscript, except for the first two, could, and should, have uncertainty ranges (like 95

- *Correlation maps (e.g. Figure 3 and 4) are done for the ensemble mean field. The correlation between individual ensemble members is relatively low, and the weather/climate signal only really emerges in the ensemble mean. Also, including this spread in a map would make the figure very hard to read, and other important features are lost.*
- *For the maps of the EOF patters (e.g Figure 5) the 2^{nd} and 3^{rd} EOFs also only emerge clearly for the ensemble mean data. For this reason, including the ensemble spread for the time series of the PCs (Figure 6) is not really feasible. And would also make the already busy figures hard to read.*
- *We do discuss the uncertainty in Section 5, but we do see the point of the Referee and we will expand this discussion to include the points we mention above, and also include a figure to exemplify the ensemble spread of temperature (see figure below including ensemble mean and spread (red) compared to DJF temperature for Greenland coastal stations (black)).*

[Figure]

Additional issues/notes

I'm a little confused about how the analog matching is working, based on figure 1. Specifically, are any adjustments made to the model-output before calculating the EOFs of δ 18 O? If not, I'm confused about how there is such fine spatial structure in the model δ 18 O, given that it has 3.5 degree grid cells. In fact, I think it would be helpful to see the outlines of the gridcells on the lower half of Figure 1. Maybe there's enough resolution there, but I found it confusing. I'm also pretty surprised about how comparable the modeled and observed δ 18 O EOFS are, they're nearly identical. I'm not particularly familiar with this region and proxy, but model-proxy EOF comparisons this similar are exceptionally rare, unless one was forced/derived from the other, and I'd be interested to learn more about this.

- *There are no adjustments made to the model output before comparing the EOFs in Figure 1. This is the whole point – that we can match the modeled patters to ice core data without tuning/calibration.*
- *The model grid is relatively course, corresponding to ~400 km at the Equator. However, since the grid gets denser at higher latitudes, there are quite a few grid points covering Greenland. To illustrate this, we follow the Referee's advise and add the grid to Figure 1 d-f.*
- *The match of the model to the patterns is indeed very good, and without this the method would not work. The good match means that the average variability of the modeled d18O is realistic on the regional scale. In general ECHAM5-wiso has been found to score very high in model-data comparisons using isotope enabled models (e.g. Steen-Larsen et al. (2017).*

Here's a suggestion that might be beyond the scope of this manuscript, but that I think is interesting: have you considered trying to assimilate different proxies for different seasons, but for the same assimilation? It would be really interesting to see what an annual reconstruction looks like where tree rings were assimilated for summer, while ice cores were simultaneously assimilated for winter – i.e., do the analog matching differently for each season but find the years that match both optimally.

- *This is certainly an interesting question, while it is beyond the scope of this study. It would require quite a lot of testing and also put strong constraints on the seasonal variability of the model and which model analogues that can be chosen. As indicated by Figure 2, there is limited co-variability between the seasons, however there is potentially some additional information on the climate variability to be gained from this approach. In the study by Tardif et al. (https://doi.org/10.5194/cp-15-1251-2019) they use seasonal proxies to reconstruct the annual variability, however there is no analysis of seasonal reconstructions in that study, so it is hard to know to which extent they are successful.*

Minor issues:
Line 7. "Reconstructs" should be "Reconstructions"
- *Corrected*

32: 18-O should have the standard superscript formatting
- *Corrected*

85: "extend" should be "extent"
- *Corrected*

328: "depended" should be "dependent"
- *Corrected*

385: "particularly" should be "particular"
- *Corrected*

Figure 7: Add some additional labels to the panels to help differentiate. It took me awhile to figure out why c and d were separated.
- *Correceted*

*References.*
*Steen-Larsen, H. C., C. Risi, M. Werner, K. Yoshimura, and V. Masson-Delmotte (2017), Evaluating the skills of isotope-enabled general circulation models against in situ atmospheric water vapor isotope observations, J. Geophys. Res. Atmos., 122, 246–263, doi:10.1002/2016JD025443.*

---

## Author Response (AR1)

Jesper Sjolte
Department of Geology
Quaternary Sciences
Lund University
Sölvegatan 12223 62 Lund
Sweden
Tel:     +46 46 222 39 92
Email:  jesper.sjolte@geol.lu.se

April 30, 2020

To: Climate of the Past

Dear Nerilie,

Please find a point-by-point response to the reviewer comments below, as well as a marked-up manuscript version of the changes in the revised version. Following your advise we have added "methodological" to the title to clarify that this is primarily a method study. The title now reads *Seasonal reconstructions coupling ice core data and an isotope enabled climate model – methodological implications of seasonality, climate modes and selection of proxy data.* We have done our best to carefully follow the reviewer comments. Following your suggestion, we have revised the manuscript to a greater extent than we outlined in our reply in the discussion. Important changes include:

- A more detailed method description, including a new Supplementary Figure S2.
- A more complete discussion and illustration of the uncertainties in our reconstruction, including a new Figure 5 and Supplementary Figure S16.
- A new section *4.3 Comparison to other millennium-length reconstructions* comparing our reconstruction to reconstructions using data completely independent from the data used in our reconstruction. This section includes new Figures 14 and 15. Figure 15 is also relevant for the discussion of uncertainties.
- A discussion of the implications of the ensemble reconstruction method on the amplitude of year-to-year variability, including aspects of the new Figures 5 and 15.

To clarify the scope of the paper we now state in the abstract and introduction that this is a method study, in addition to the changed title.

We hope our efforts to improve the manuscript bring it up to a satisfactory level.

Sincerely,

Jesper Sjolte

On behalf of all authors.

*Reply to Referee #1. Author comments in blue italics.*

Anonymous Referee #1

The manuscript presents several reconstructions of climate fields in the North Atlantic over the last centuries. The reconstructions are based on proxy data on the one hand (oxygen isotope concentrations in Greenland ice cores and on European dendroclimatological data ) and on climate simulations with a isotope enabled climate model. Both types of data are combined applying the analog method. The manuscript is mainly focused on testing the sensitivity of the reconstructions on the number of ice cores, on the definitions of the target seasons. The manuscript is less focused on the physical results and the implications of these reconstructions. It is therefore rather technical manuscript. There are almost no results or dicussion on the physical mechanisms or past climate variations. This is in principle fine, but the readers should be made aware of this early on in the abstract and in the introduction. In my opinion, the manuscript addresses interesting issues. For instance, the use isotopic data from ice cores in combination with climate simulations to reconstruct the atmospheric circulation is an interesting idea. However, being this a more technical manuscript, the description of the methods applied leaves many open questions for the reader. This description should be considerably improved. At some points it is so unclear that I had doubts about the correct application of the method, although I hope that it is in the end correct.

- *We are grateful for the very detailed and helpful comments by the reviewer, which has helped us to produce a much improved revised manuscript. The method used in this manuscript was first published in Sjolte et al. (2018). Therefore some aspects of the description of the methodology is less detailed. This is clearly not the right way to do it and we have worked on clarifying technical details and the motivation of our approach. We also agree the this is mainly a method paper. This is stated clearly in the abstract and introduction of the revised manuscript, and we have also changed the title.*

Main points:
1. One main concern is the limited methodological description. The authors apply the analog method after a pre-filtering by Principal Component Analysis, but it is not cleat how this pre-filtering is actually conducted. Important questions that may impact the results :
1.1 Are the PCs calculated from the covariance of correlation matrix?

- *The PCs are calculated from the covariance matrix. This is specified in the revised manuscript*

1.2 in Equation 1, are the PCS normalized to unit variance or has each PC the platitude representing the corresponding explained variance. This is important because this point represents two options: all PCs are equally weighted for the calculation of the distance of the analogues, or each PCs is weighted according to the varianve it represents. The selected analogues are different depending on the option chosen.

- *The PCs are normalized to unit variance. This is specified in the revised manuscript. The approach of fitting the PCs is motivated by that we want to capture as much of the regional signal from ice cores as possible, but not overfit noise at individual sites. Fitting normalized PCs allows us to capture the 'average' signal at all the ice core sites, while constraining this signal with the regional variability between the cores. This is now described in the revised manuscript.*

1.3 More importantly, for a correct application of the method, the EOFs patterns
(derived from ice-core records and from model grid cells) associated to each PC
in equation 1 must be the same . Otherwise, the PC-coordinates PC_{model} and
PC_{icecores} ) would not be linked to the same EOFs and therefore would represent
coordinates in different subspaces. The calculated distances would not be meaningful.
It may happen that the patterns of EOF_{model} and EOF_{icecores} are very similar,
in which case this problem would be minor. But this is not guaranteed. A way to ensure that the PCs
in equation 1 refer to the same EOF patterns is to use, for instance,
EOF_{model} for both data sets and calculate PC_{icecores} by projecting the ice core
anomalies onto EOF_{model} . (Or viceversa, use EOF_{icecores} for both data sets).
This is the point that most strongly worries me. If EOF_{model} and EOF_{icecore}
patterns are really different the whole application of the method is not correct, and all
results should be re-calculated.

- *The EOFs of the model output and ice core data are indeed very similar. The map in Figure 1 is to some extent showing this, although it can be hard to determine how close from a map. As shown in Sjolte et al. (2018) we actually use an additional step to test if the selected model analogues fit the original ice core data (so, not the PCs), where we extract the reconstructed ensemble mean model d18O and correlate it to the ice core data. This step tests if the matching of the PCs works. We include a description of this step of the reconstruction method in the revision, and we have illustrated this in a new Supplementary Figure S2 showing the correlation between reconstructed d18O and ice core d18O for summer, winter and annual data.*

2. Another unclear methodological point is how the dendro data are included for the
reconstruction. Here, I cannot make any useful suggestion because the authors ap-
proach remains unclear to me. This needs to be much better explained:
2.1 Are the analogues searched using ice-core and dendro data simultaneously, i.e. a
12-month-long model analog have to be close to the icecore data in the target season
and close to the dendro-reconstructed summer temperatures in the summer season?.
If yes, how is the EOF filtering implemented here? How many 'temperature' EOFs are
used.
or
2.2 Do the selected analogues (using ice core data) undergo a secondary selection
procedure targeting the dendro-reconstructed temperatures ?

- *It is a secondary selection as stated in L145 "we sort the 39 existing ensemble members based on the ice core selection". The test with the tree-ring data is to see if we can constrain the temperature further. This is not meant as a final reconstruction, but as a test of the common signal between Greenland and European proxy data. As the results show there is quite some common signal for ~20 of the analogues out of the 39 pre-selected analogues based on ice core data. The purpose and technical details connected to the use of tree-ring data has be clarified both in terms of methods and motivation.*

2.3 In both cases, are the distances to the dendro-data and the distances to the ice
core data equally weighted ? How is this implemented if the number of EOFs for each
type of data sets is presumably not the same.

- *See reply above.*

3. The validation of the results is essentially made by calculating the correlation be-
tween reconstructions and 20CR reanalysis. However, the amplitude of the reconstruc-
tions is not validated. This may be important because the amplitude of reconstructed
variability may depend on the number of analogs selected: best-analog-selection (just
one analogue) will roughly produce the same amplitude of variations, although the

validation correlation will be lower; in contrast, using the mean of a larger number of analogues subdues the variability, and this effect can be substantial when using 39 (?) analogues. There is an unavoidable trade-off between better correlations and more realistic amplitudes, as shown in Gomez-Navarro et al. Pseudo-proxy tests of the analogue method to reconstruct spatially resolved global temperature during the Common Era, Clim. Past, 13, 629–648, https://doi.org/10.5194/cp-13-629-2017, 2017 ).

- *This point is discussed by Sjolte et al. (2018) in terms of the amplitude of d18O for the method used here. There is a trade off, and some of the amplitude of the signal is smoothed out when using the ensemble approach. Comparing to the d18O amplitude in the ice cores data this is a minor effect, and some of the amplitude in the ice core data is in fact depositional noise, which we don't want to fit the model data to.*
- *We do show the amplitude compared to SST data, where it is quite realistic for winter and annual variability. The underestimation of the SST amplitude for summer has more to do with the model biases and seasonal climate/proxy differences already discussed in the manuscript.*
- *We discuss this point further in a new paragraph in Discussion and Conclusions (L435-447). We also include new Figures 5 and 15 relevant for this discussion, and refer to Gomez-Navarro et al. (2017).*

Particular points:

4. We test a range of climate reconstructs varying the definition of the seasons
climate reconstructions
- *Corrected*

5. The abstract does not mention the reconstruction method at all, despite the manuscript being essentially methodological in nature.
- *Corrected*

6. best captured when defining the season December-February
the season as December-February
- *Corrected*

7. line 10 best captured when defining the season December-February due to the dominance of large scale patterns, while for summer the weaker, albeit more strongly auto-correlated, variability is better captured using a longer season of May-
This sentence becomes clearer later in the manuscript. Here, I would suggest to improve its clarity, for instance, using 'more persistent in time' instead of autocorrelated.
- *Corrected*

8. One point that sets the study by Sjolte et al. (2018) apart from the other studies mentioned in this section, is the use of
delete comma after section
- *Corrected*

9. line 145 For the summer reconstructions also using tree-ring data we sort the 39 existing ensemble members
I am rather confused by this sentence. The number 39 is mentioned here for the first time, if I am not mistaken. What are these ensemble members? are they the analogs previously selected targeting the ice-core data ?
- *See reply to major point 2.2*

10. line 155 In this study we follow the convention of using the term PCs for the time series of the main modes of variability, while using the term EOFs for the spatial patterns of the modes. The method of Ebisuzaki (1997) is used to calculate the significance when
this sentence should appear before equation 1, at the very least
  - *Corrected*

11. line 163 A key factor in how well seasonal climate reconstructions can represent climate itself, is the auto-correlation structure of atmospheric
climate itself ? I guess the authors mean to what extent can seasonal proxy data represent annual means ?
  - *Corrected. It should have been "A key factor*
  - *in how well seasonal proxy data can represent climate variability, is the sub-seasonal auto-correlation structure of atmospheric variability. " the second part of the sentence is important for context, and we added "sub-seasonal" to be more precise.*

12. line 167 Figure 2 shows the monthly auto-correlation of each month of the PC-based NAO calculated from the 20CR. These figures show that during the cold season the
Which is the PC that represents the NAO ? Here, it is assumed that, for each month the leading PC is the NAO. In summer this is not always the case, and it depends on the geographical region selected to conduct the PCA.
  - *We checked the patterns and the leading PC is NAO for all months. However, as we write there is little consistency between the months of the secondary PCs.*

13. line 180
circulation modes. We do this by performing monthly reconstructions for pressure and evaluating the resulting main modes of circulation against the modes of the 20CR. This is done using the same method as for the seasonal reconstructions,
what does 'evaluating the modes' mean here ?. The spatial patterns (EOFs) resulting from an analog reconstructions can be very realistic irrespective of the skill of the analog method. The analog method is just a resampling from a data pool. A random resampling of SLP from the 20CR reanalysis or from a model run will produce the same EOFs as the orginal data set, so even if the analog is wrongly implemented, the resulting EOFs may look correct. This is different for a temporal validation, e.g. correlation between reconstructions and observations, where the skill of the analog selection is critical. The authors should be here more specific.
  - *The resampling changes the major modes, so the modes do not simply come from the model, and the modes are also reshuffled when the variability is resampled. We write this in L273-276 of the original manuscript. We have further clarified this section in the revision.*

Anonymous Referee #2

The manuscript by Sjolte et al. investigates a new innovation in the rapidly developing field of paleoclimate data assimilation. Specifically, they investigate the potential of reconstructing seasonal fields using subannually resolved ice-core (and to a lesser extent, tree ring) data from the North Atlantic. The manuscript is well-written, well-illustrated and generally well-organized, and the results are interesting, and suited to Climate of the Past. I do however, have a few concerns and and suggested additions to the manuscript that I'd like to see addressed.

- *We thank the reviewer for the positive comments and interest in out manuscript, as well as the detailed comments which helped us greatly improve the manuscript.*

Major Issues

In general, in my opinion, the primary weakness of the manuscript is that the exploration of the reanalyses is rather limited. For example, in the authors subdivide the ice cores into a group of 8 that extends from 1241-1970 and a larger group that is shorter (1777-1970). However the reconstructions are only analyzed in the context of instrumental data. No results from prior to 1850 are shown in the manuscript or supplement, except for figure S3, which is specifically focused on the tree ring sites. In evaluating this technical approach, it is important for readers to be able to see how the longer term variability compares to other reconstructions from the region, and to consider and discuss how the seasonal assimilations affect long-term variability, and the potential climatic implications of that. Given that this approach creates a field reconstruction; these results could be compared to regional temperature reconstructions, NAO reconstructions, and more, and give the readers a better sense of how this approach compares with previous efforts.

- *We agree that an in-depth comparison to other reconstructions of the longer term variability would be very interesting. For the DJF NAO reconstruction covering 1241-1970 this is already done by Sjolte et al. (2018). The purpose of this paper is to test detailed aspects of seasonal variability and which factors affect the skill of seasonal reconstructions. These tests are only possible with observation-based data where we have full control on the time scale and seasonality. Furthermore, a full-scale comparison to the long term variability of other reconstructions is a whole study in itself, and this manuscript is already crowded by many figures and results. As also pointed out by Referee #1 the scope of this study should be better defined. We have done so in the revised manuscript, and also included a comparison to other reconstructions limited to two reconstructions that are completely independent from ours. This is shown in a new section 4.3, including two new figures (Figure 14 and 15).*

At present, the evaluation of the results is restricted to spatial comparison of the first three PCs with instrumental data, temporal comparison of the same thing. I was glad to see SSTs averaged and compared to instrumental data, but feel like the comparison was ultimately very limited.

- *Please note that we also compare to the station-based NAO (Figure 7c and 7d), as well as temperature data from Greenland and Iceland (Table 4). Again, the motivation for using only observation-based data is that this is the only data where there is no uncertainly with respect to sub-annual temporal resolution. We would like to point out that the correlation*

*maps (Figure 3, 4, S7, S8, S9) are also temporal comparisons, although illustrating the spatial extent of the skill.*

The other major weakness of the manuscript, that I believe should be able to address, was the representation of uncertainty. The methodological approach to uncertainty quantification; an ensemble based approach, is reasonable. I was disappointed however that the results were not presented in the manuscript. Every figure in the manuscript, except for the first two, could, and should, have uncertainty ranges (like 95

- *We have expanded the discussion of the aspects of uncertainty with new text (L421-429) including the new Figure 5 showing Greenland coastal temperatures, including the ensemble spread and RMSE. In the new Figure 15 (NAO reconstruction comparison) we also show ensemble spread and RMSE and include this in the discussion of uncertainty. In summary we argue that the ensemble spread is a good measure of uncertainty, as it is comparable to the RMSE with respect to observations.*
- *In a new Supplementary Figure S16 we show the normalized RMSE together with the map for reconstructed DJF SLP, T2m and SST correlated against the 20CR. This is an alternative way of showing the skill, taking into account the amplitude of the variability. However, we limit our selves to showing it for this subset of the reconstruction as an example, instead of crowding the study with even more figures. We also discuss Figure S16 in the new text for section 5, Discussion and conclusions.*

Additional issues/notes
I'm a little confused about how the analog matching is working, based on figure 1. Specifically, are any adjustments made to the model-output before calculating the EOFs of $\delta 18 O$? If not, I'm confused about how there is such fine spatial structure in the model $\delta 18 O$, given that it has 3.5 degree grid cells. In fact, I think it would be helpful to see the outlines of the gridcells on the lower half of Figure 1. Maybe there's enough resolution there, but I found it confusing. I'm also pretty surprised about how comparable the modeled and observed $\delta 18 O$ EOFS are, they're nearly identical. I'm not particularly familiar with this region and proxy, but model-proxy EOF comparisons this similar are exceptionally rare, unless one was forced/derived from the other, and I'd be interested to learn more about this.

- *There are no adjustments made to the model output before comparing the EOFs in Figure 1. This is the whole point – that we can match the modeled patters to ice core data without tuning/calibration.*
- *The model grid is relatively course, corresponding to ~400 km at the Equator. However, since the grid gets denser at higher latitudes, there are quite a few grid points covering Greenland. To illustrate this, we follow the Referee's advise and have added the grid to Figure 1 d-f.*
- *The match of the model to the patterns is indeed very good, and without this the method would not work. The good match means that the average variability of the modeled d18O is realistic on the regional scale. In general ECHAM5-wiso has been found to score very high in model-data comparisons using isotope enabled models (e.g. Steen-Larsen et al. (2017).*
- *Following the comments of both reviewers we have provided a more detailed method description in the revised manuscript.*

Here's a suggestion that might be beyond the scope of this manuscript, but that I think is interesting: have you considered trying to assimilate different proxies for different seasons, but for the same assimilation? It would be really interesting to see what an annual reconstruction looks like where tree rings were assimilated for summer, while ice cores were simultaneously assimilated for winter – i.e., do the analog matching differently for each season but find the years that match both optimally.

- *This is certainly an interesting question, while it is beyond the scope of this study. It would require quite a lot of testing and also put strong constraints on the seasonal variability of the model and which model analogues that can be chosen. As indicated by Figure 2, there is limited co-variability between the seasons, however there is potentially some additional information on the climate variability to be gained from this approach. In the study by Tardif et al. (*https://doi.org/10.5194/cp-15-1251-2019*) they use seasonal proxies to reconstruct the annual variability, however there is no analysis of seasonal reconstructions in that study, so it is hard to know to which extent they are successful.*

Minor issues:
Line 7. "Reconstructs" should be "Reconstructions"
- *Corrected*

32: 18-O should have the standard superscript formatting
- *Corrected*

85: "extend" should be "extent"
- *Corrected*

328: "depended" should be "dependent"
- *Corrected*

385: "particularly" should be "particular"
- *Corrected*

Figure 7: Add some additional labels to the panels to help differentiate. It took me awhile to figure out why c and d were separated.
- *Corrected*

*References.*
*Steen-Larsen, H. C., C. Risi, M. Werner, K. Yoshimura, and V. Masson-Delmotte (2017), Evaluating the skills of isotope-enabled general circulation models against in situ atmospheric water vapor isotope observations, J. Geophys. Res. Atmos., 122, 246–263, doi:10.1002/2016JD025443.*

[revised manuscript text omitted]

---

## Author Response (AR2)

Jesper Sjolte Department of Geology Quaternary Sciences Lund University Sölvegatan 12223 62 Lund Sweden Tel: +46 46 222 39 92 Email: jesper.sjolte@geol.lu.se

To: Climate of the Past

Dear Nerilie,

Thank you very much for the helpful and constructive comments. Please find a point-by-point response to your comments below, as well as a version of the manuscript marked-up with the changes in the revised version. We have for the most part followed your suggestions closely, which has made the manuscript easier to follow.

Sincerely, ~ Solo Jesper Sjolte

On behalf of all authors.

July 1, 2020

**Reply to comments by the editor**

Please find author reply in blue italics below.

**Editor Decision: Publish subject to minor revisions (review by editor)** (24 Jun 2020) by Nerilie Abram

Comments to the Author:

Dear Jesper Sjolte and co-authors,

Thank you for responding to the reviewer comments and submitting a revised version of your manuscript to Climate of the Past. I am satisfied that the reviewer comments have been addressed well and I am pleased to accept your manuscript for publication with minor revisions. Please address the following editor comments in preparing a revised manuscript:

\*Please carry out a thorough proof reading of the article to fix typographical errors in the text

• Done

\*Figure 14: I'm not convinced that the pre-1500 comparison is reliable. The scaling of this lower resolution data does not appear to be appropriate and so it would seem better to only compare the reconstructions from 1500 onwards.

• We have excluded the pre-1500 comparison and explained why in the text.

\*Figure 14 and 15: It would be helpful to have a consistent approach to the moving correlations. e.g. in figure 14 all of the moving correlations are shown but there is no indication of significance. In figure 15 the moving correlations are only shown where they are significant.

• The indication of significance is now the same in Figure 14 and 15, and done in a clearer way.

\*Figure 3, Figure 4 and Table 3: I don't think that it is appropriate to give mean correlations only for significant correlations. This doesn't give an accurate reflection of the skill of the reconstructions, and will also be affected by the relative size of regions of positive and negative correlations. Suggest restricting this only to reporting the number of grid cells with significant correlations. Alternately, reporting the maximum correlation values would be more informative.

• We now indicate the maximum correlation instead of mean correlation in Figure 3, 4, Supplementary Figure S7, S8, S9 and S16 as well as in Table 3.

\*Table 3: The arrangement of Table 3 is difficult to follow. Please group summer/JJA and winter/DJF categories near to each other. I've already suggested not reporting the mean of significant correlations, but if these are kept or swapped for reporting maximum correlations please also group the correlations and # grid cell information for each parameter together.

**• The grouping of seasons and parameters are now done as you suggest, with the mean correlation replaced by maximum correlation.**

\*Please consider archiving the code associated with this paper to aid with promoting FAIR data practices. For example, as this paper is about a reconstruction method it would be helpful for the

community for the code associated with the method to be archived and made accessible on a platform like github.

• We appreciated this suggestion, however, the code is very specific to the formatting of the input data, and the part of the code for the reconstruction itself is a very small part of the script. We judge that anyone interested in more details would have to contact the corresponding author anyway if information is needed beyond what is given in the method section. We have made a statement on code and data availability at the end of the paper.

Please let me know if any of this isn't clear. I look forward to receiving your revised manuscript.

Sincerely, Nerilie

**Seasonal reconstructions coupling ice core data and an isotope enabled climate model – methodological implications of seasonality, climate modes and selection of proxy data**

Jesper Sjolte1, Florian Adolphi1,3, Bo M. Vinther4, Raimund Muscheler1, Christophe Sturm2, Martin Werner5, and Gerrit Lohmann5

[revised manuscript text omitted]